# ON THE HÖLDER STABILITY OF MULTISET AND GRAPH NEURAL NETWORKS

**Yair Davidson**[1] **Nadav Dym**[1,2]

[1] Department of Computer Science - Technion, Haifa, Israel
[2] Department of Mathematics - Technion, Haifa, Israel

## ABSTRACT

Extensive research efforts have been put into characterizing and constructing maximally separating multiset and graph neural networks. However, recent empirical evidence suggests the notion of separation itself doesn't capture several interesting phenomena. On the one hand, the quality of this separation may be very weak, to the extent that the embeddings of "separable" objects might even be considered identical when using fixed finite precision. On the other hand, architectures which aren't capable of separation in theory, somehow achieve separation when taking the network to be wide enough.

In this work, we address both of these issues, by proposing a novel pair-wise separation quality analysis framework which is based on an adaptation of Lipschitz and Hölder stability to parametric functions. The proposed framework, which we name *Hölder in expectation*, allows for separation quality analysis, without restricting the analysis to embeddings that can separate all the input space simultaneously. We prove that common sum-based models are lower-Hölder in expectation, with an exponent that decays rapidly with the network's depth. Our analysis leads to adversarial examples of graphs which can be separated by three 1-WL iterations, but cannot be separated in practice by standard maximally powerful Message Passing Neural Networks (MPNNs). To remedy this, we propose two novel MPNNs with improved separation quality, one of which is lower Lipschitz in expectation. We show these MPNNs can easily classify our adversarial examples, and compare favorably with standard MPNNs on standard graph learning tasks.

## 1 INTRODUCTION

Motivated by a multitude of applications, including molecular systems (Gilmer et al., 2017), social networks (Borisyuk et al., 2024), recommendation systems (Gao et al., 2023) and more, permutation invariant deep learning for both multisets and graphs have gained increasing interest in recent years. This in turn has inspired several theoretical works analyzing common permutation invariant models, and their expressive power and limitations.

For multiset data, it is known that simple summation-based multiset functions are injective, and as a result, can approximate all continuous functions on multisets (Zaheer et al., 2017). These results have been discussed and strengthened in many different recent publications (Wagstaff et al., 2019; Amir et al., 2023; Tabaghi & Wang, 2024; Wang et al., 2024).

For *graph neural networks* (GNNs) the situation is more delicate, as all known graph neural networks with polynomial complexity have limited expressive power. Our focus in this paper will be on the Message Passing Neural Network (MPNN) (Gilmer et al., 2017) paradigm, which includes a variety of popular GNNs (Xu et al., 2019; Gilmer et al., 2017; Kipf & Welling, 2017). In their seminal works, Xu et al. (2019); Morris et al. (2019) analyze the separation capabilities of MPNNs, showing that an MPNN $f$ can separate two graphs $G$ and $H$ (i.e., $f(G) \neq f(H)$) only if the Weisfeiler-Lehman (WL) isomorphism test (Weisfeiler & Lehman, 1968) can also do so. Accordingly, maximally expressive MPNNs are those which are able to separate all graph pairs which are WL-separable. They further show that MPNNs which employ injective multiset functions are maximally expressive.

The theoretical ability of a permutation invariant network to separate a pair of objects (multi-sets/graphs) is a necessary condition for all learning tasks which require such separation, e.g., binary classification tasks where the two objects have opposite labels. However, while current theory ensures separation, it tells us little about the separation quality, so that embeddings of "separable" objects may be extremely similar, to the extent that in some cases, graphs which can be theoretically separated are completely identical on a computer with fixed finite precision. This pheonomena was observed for graphs with analytic activations and very small width, see (Bravo et al., 2024), figure 2 top.

In a non-parametric setting, separation quality of a function $f$ can be studied via bi-Hölder stability guarantees: for metric spaces $(X, d_X)$ and $(Y, d_Y)$, a function $f : X \to Y$ is $\beta$ *upper-Hölder* and $\alpha$ *lower-Hölder* if there exist some positive constants $c, C$ such that

$$c d_X(x, x')^\alpha \le d_Y(f(x), f(x')) \le C d_X(x, x')^\beta, \forall x, x' \in X.$$

The upper and lower Hölder conditions guarantee that embedding distances in $Y$ will not be much larger, or much smaller, than distances in the original space $X$ (in our case a space of multisets or graphs). A function $f$ will have higher separation quality the closer the Hölder exponents are to one. When $\beta = 1$ we say $f$ is (upper) Lipschitz, and when $\alpha = 1$ we say that $f$ is lower Lipschitz.

Bi-Lipschitz stability analysis is a central topic in the study of frames Balan (1997); Alexeev et al. (2012), phase retrieval Bandeira et al. (2014); Cheng et al. (2021), and several group-invariant learning scenarios Cahill et al. (2020; 2024b). For multisets, Bi-Lipschitz stability results are known for multiset-functions based on max-filters Cahill et al. (2024c) or sorting Balan et al. (2022). However, this is not the case for more common sum-based multiset functions: Amir et al. (2023) showed that *Relu-sum* multiset functions, which are based on summation of point-wise applied ReLU networks (see (1) below) are never even injective. Multiset functions which use *smooth activations* instead, are injective but not lower-Lipschitz. Accordingly, a remaining challenge, which we will address in this paper, is characterizing the lower-Hölder stability of sum-based multiset functions.

For MPNN architectures for graphs, there are even less stability results. The upper Lipschitz-ness of the MPNN GIN (Xu et al., 2019) and similar architectures was established in (Chuang & Jegelka, 2022), however lower Lipschitz or Hölder guarantees have not been addressed to date. Indeed, a recent survey (Morris et al., 2024) lists bi-Lipschitz guarantees for MPNNs as one of the future challenges for theoretical GNN research. We will address this challenge in this paper as well.

As a first step for our stability analysis, we need to establish how to extend notions of Hölder stability to parametric functions. One simple approach is requiring the parametric function to be *uniformly* Hölder, with the same exponent and constant regardless of the parameter choice. Indeed, we will show that all parameteric functions we consider in this paper are uniformly *upper*-Lipschitz.

In contrast, we believe the notion of uniform *lower*-Hölder to be too stringent. For example, as mentioned above, multiset networks based on ReLU activations are never injective, and so can't be uniformly lower-Hölder. Nonetheless, numerical estimates of the stability of such networks show that wide ReLU networks have comparable or even better stability than injective smooth-activation-based multiset functions (see Amir et al. (2023) Figure 2). Accordingly, we resort to a probabilistic framework of *Hölder stability in expectation*, as presented in section 2.

**Main results: multisets**   With respect to our new relaxed notion of expected lower-Hölder, we show that ReLU-sum multiset networks have an expected lower-Hölder exponent of $\alpha = 3/2$. Surprisingly, while smooth activations lead to injectivity, we find that their expected Hölder exponent is much worse: at best it is equal to the maximal number of elements in the multiset data, $\alpha \ge n$. This scenario is summarized in Figure 1: For a given pair of multisets, smooth activations will separate even with a very small network width, but the separation quality can be very poor. In contrast, shallow Relu-sum networks may not attain separation. However, as width increases separation will be obtained (with high probability), and the expected quality of separation will surpass the quality attained with smooth activations.

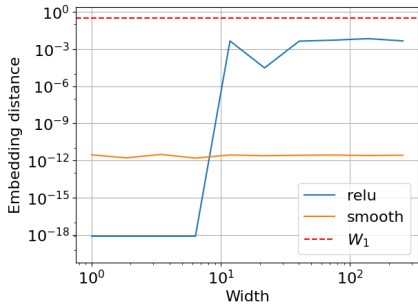

Figure 1: Separation quality on a single adversarial multiset-pair constructed as described in appendix G.3.

The relatively moderate exponent $\alpha = 3/2$ of ReLU networks is guaranteed only when the range of the bias of the network contains the range of the multiset features, an assumption which may be problematic to fulfill in practice. To address this, we suggest an *Adaptive ReLU* summation network, where the bias is adapted to the values in the multiset. This network is guaranteed the same $3/2$ exponent while overcoming the range issue.

Finally, we consider multiset functions based on linear functions and column-wise sorting. These were shown to be bi-Lipschitz, when wide enough, in (Balan et al., 2022). We show that they are lower Lipschitz *in expectation* even with a width of 1.

**Main Results: MPNNs** MPNNs are constructed upon multiple application of multiset functions to node neighborhoods. We consider four MPNNs based on the four multiset functions we analyzed: SortMPNN, ReluMPNN, SmoothMPNN and AdaptMPNN. We show that SortMPNN is lower-Lipschitz in expectation, even with a width of 1. SortMPNN is thus the first MPNN with both upper and lower Lipschitz guarantees. In contrast, we show that ReluMPNN and SmoothMPNN are only lower-Hölder, with an exponent that deteriorates as the MPNN depth grows. The exponent bounds of both the graph and multiset models appear in table 1.

Table 1: Summary of lower-Hölder exponent bounds for multiset (top) and graph (bottom) models.

| Model | bounds |
|---|---|
| relu sum | $\alpha = 3/2$ |
| adaptive relu | $\alpha = 3/2$ |
| smooth sum | $n \leq \alpha$ |
| **sort based** | $\alpha = 1$ |
| ReluMPNN | $1 + \frac{K+1}{2} \leq \alpha$ |
| SmoothMPNN | $2^{K+1} \leq \alpha$ |
| **SortMPNN** | $\alpha = 1$ |

Our analysis provides an adversarial example illustrating the deterioration of the expected lower-Hölder exponent of sum-based MPNNs as the depth increases. We show in Table 2 that sum-based MPNNs fail to learn a simple binary classification task on such data, while SortMPNN and AdaptMPNN handle this task easily. We also provide experiments on several graph datasets which show that SortMPNN often outperforms standard MPNNs on non-adversarial data, and is more robust to reduction in model size.

**1.0.1 Notation** $S^{d-1}$ denotes the unit sphere in $\mathbb{R}^d$. The notation '$a \sim S^{d-1}$ and $b \sim [-B, B]$' implies that the distribution on $a$ (respectively $b$) is taken to be uniform on $S^{d-1}$ (respectively $[-B, B]$), and that $a$ and $b$ are drawn independently. $x \cdot y$ denotes the inner produce of $x, y$.

## 2 GENERAL FRAMEWORK

In this section, we present the framework for analyzing separation quality of parametric functions. We begin by extending the notion of Hölder stability to a family of parametric functions.

### 2.1 HÖLDER STABILITY FOR PARAMETRIC FUNCTIONS

Let $(X, d_X)$ and $(Y, d_Y)$ be psuedo-metric[1] spaces. Let $W$ be some set of parameters, and let $f(x; w) : X \times W \to Y$ be a parametric function. We say that $f(x; w)$ is *uniformly* Lipschitz with constant $L > 0$, if for all $w \in W$, the function $x \mapsto f(x; w)$ is $L$-Lipschitz. This definition can naturally be extended to other upper-Hölder exponents, but this will not be necessary as the parametric functions we discuss in this paper will all end up being uniformly Lipschitz. Similarly, we could define a notion of uniform lower-Hölder, but as discussed in this introduction, this notion is too stringent for the problems we are discussing, and so we introduce the alternative notion of lower-Hölder *in expectation*:

**Definition 2.1.** Let $\alpha > 0, p \geq 1$. Let $(X, d_X)$ and $(Y, d_Y)$ be psuedo-metric spaces, and let $(W, \mathcal{B}, \mu)$ be a probability space. Let $f(x; w) : X \times W \to Y$ be a parametric function, so that for every fixed $x \in X$ the function $f(x; \cdot)$ is measurable.

We say that $f$ is $\alpha$ lower-Hölder *in expectation* if there exists a strictly positive $c$ such that

$$c^p \leq \mathbb{E}_{w \sim \mu} \left\{ \frac{d_Y(f(x; w), f(x'; w))}{d_X(x, x')^{\alpha}} \right\}^p, \quad \forall (x, x') \in X \times X, \text{ with } d(x, x') > 0$$

---

[1]A psuedo metric $d_X$ differs from a standard metric only in that the $d_X(x_1, x_2) = 0$ is possible even if $x_1 \neq x_2$.

A few remarks on this definition are in order. Firstly, we note that if the set of $w$ for which $f(x; w)$ is $\alpha$ lower-Hölder has positive probability, then $f(x; w)$ will be $\alpha$ lower-Hölder in expectation. The opposite is not true: Relu-sum networks for multisets are never injective, and hence never lower-Hölder, but we will show that they are lower-Hölder in expectation. This is possible since it is defined in a pairwise sense, similar to the separation analysis in Morris et al. (2019).

Next, we note that our definition takes an expectation over the parameters, but requires uniform boundedness over $x, x'$ pairs. This choice was made since data distribution is unknown, while the parameter distribution at initialization can be chosen by the algorithm. Furthermore, we conjecture that bad separation at initialization will be difficult to overcome during training. Supporting evidence for this claim will be shown later on (see table 2), where models with a large lower-Hölder in expectation exponent, fail to learn an adversarial binary classification task.

Thirdly, we note that Hölder in expectation exponents that are closer to one will be regarded as having higher separation quality. Fourthly, $p$ in the definition refers to the $\ell_p$ norm chosen for the feature spaces. Results stated in the introduction are for the default $p = 2$.

Finally, while the expected value across random parameters is informative as is, the variance can potentially have a strong effect on the separation in practice. Luckily, our analysis will typically apply for models $f(x; w)$ of width 1, and models with width $W$ can be viewed as stacking of $W$ multiple model instances. Wider models will have the same expected distortion as the width one model, but the variance will converge to zero, in a rate proportional to $1/W$ For more details see Appendix B.1.

We now proceed to analyze Hölder stability for parametric models on multisets.

## 3    Multiset Hölder stability

**Standing assumptions:** Throughout the rest of this section, we will say that $(n, d, p, \Omega, z)$ satisfy our standing assumptions, if $n, d$ are natural numbers, $p \geq 1$, the set $\Omega$ is a compact subset of $\mathbb{R}^d$, and $z$ will be a point in $\mathbb{R}^d \setminus \Omega$.

A *multiset* $\{\!\{x_1, \ldots, x_k\}\!\}$ is a collection of $n$ unordered elements where (unlike sets) repetitions are allowed. We denote by $\mathcal{S}_{\leq n}(\Omega)$ and $\mathcal{S}_{=n}(\Omega)$ the space of all multisets with at most $n$ elements (respectively, exactly $n$ elements), which reside in $\Omega$.

A popular choice of a metric on $\mathcal{S}_{=n}(\Omega)$ is the Wasserstein metric

$$W_1(\{\!\{x_1, \ldots, x_n\}\!\}, \{\!\{y_1, \ldots, y_n\}\!\}) = \min_{\tau \in S_n} \sum_{j=1}^{n} \|x_j - y_{\tau(j)}\|_1$$

Following (Chuang & Jegelka, 2022), we extend the Wasserstein metric to the space $\mathcal{S}_{\leq n}(\Omega)$ by the *augmentation* map which adds the vector $z$ to a given multiset until it is of maximal size $n$:

$$\rho_{(z)}\left(\{\!\{x_1, \ldots, x_k\}\!\}\right) = \{\!\{x_1, \ldots, x_k, x_{k+1} = z, \ldots x_n = z\}\!\}$$

This induces the *augmented Wasserstein* metric $W_1^z$ on $\mathcal{S}_{\leq n}(\Omega)$,

$$W_1^z(S_1, S_2) = W_1(\rho_{(z)}(S_1), \rho_{(z)}(S_2)), \quad S_1, S_2 \in \mathcal{S}_{\leq n}(\Omega)$$

Now that we have introduced the augmented Wasserstein as the metric we will use for multisets, we turn to analyze the multiset neural networks we are interested in.

A popular building block in architectures for multisets (Zaheer et al., 2017), as well as MPNNs (Xu et al., 2019; Schütt et al., 2017; Gilmer et al., 2017), is based on summation of element-wise neural network application. For a given activation $\sigma : \mathbb{R} \to \mathbb{R}$, we define a parametric function $m_\sigma$ on multisets in $\mathcal{S}_{\leq n}(\Omega)$, with parameters $(a, b) \in \mathbb{R}^d \oplus \mathbb{R}$, by

$$m_\sigma(\{\!\{x_1, \ldots, x_r\}\!\}; a, b) = \sum_{i=1}^{r} \sigma(a \cdot x_i - b). \tag{1}$$

We note that we focus on $m_\sigma$ with scalar outputs, since the variant with vector output has the same Hölder properties, as discussed in the end of Section 2.

**3.0.1 ReLU summation** We first consider the expected Hölder stability of $m_\sigma$, when $\sigma = \text{ReLU}$:

**Theorem 3.1.** *For $(n, d, p, \Omega, z)$ satisfying our standing assumptions, assume that $a \sim S^{d-1}$ and $b \sim [-B, B]$. Then $m_{ReLU}(\cdot; a, b)$ is uniformly Lipschitz. Moreover,*

1. *$m_{ReLU}(\cdot; a, b)$ is not $\alpha$ lower-Hölder in expectation for any $\alpha < \frac{p+1}{p}$.*

2. *If $\|x\| < B$ for all $x \in \Omega$, then $m_{ReLU}(\cdot; a, b)$ is $\frac{p+1}{p}$ lower-Hölder in expectation.*

**Proof idea: the $\pm\epsilon$ example.** The following example, which we nickname the $\pm\epsilon$ example, gives a good intuition for the Hölder behavior of $m_{\text{ReLU}}$. Denote $X_\epsilon = \{\!\{-\epsilon, \epsilon\}\!\}$ and $X_{2\epsilon} = \{\!\{-2\epsilon, 2\epsilon\}\!\}$. The Wasserstein distance between these two multisets is proportional to $\epsilon$. Now, let us consider the images of these multisets under $m_{\text{ReLU}}$, where we assume for simplicity that $d = 1$ and $a = 1$. Note that when $b > 2\epsilon$ we will get $ReLU(x - b) = 0$ for all $x$ in either sets, and so $m_{\text{ReLU}}(X_{2\epsilon}; 1, b) = m_{\text{ReLU}}(X_\epsilon; 1, b)$. We will get a similar results when $b \leq -2\epsilon$. In this case, for every $x \geq -2\epsilon$ we have $\text{ReLU}(x - b) = x - b$. Therefore, we will obtain that

$$m_{\text{ReLU}}(X_{2\epsilon}; 1, b) = (-2\epsilon - b) + (2\epsilon - b) = (-\epsilon - b) + (\epsilon - b) = m_{\text{ReLU}}(X_\epsilon; 1, b), \quad \forall b < -2\epsilon.$$

We conclude that $|m_{\text{ReLU}}(X_{2\epsilon}; 1, b) - m_{\text{ReLU}}(X_\epsilon; 1, b)|^p$ is zero for all $b$ outside the interval $(-2\epsilon, 2\epsilon)$. Inside this interval of length $4\epsilon$, we will typically have $|m_{\text{ReLU}}(X_{2\epsilon}; 1, b) - m_{\text{ReLU}}(X_\epsilon; 1, b)|^p \sim \epsilon^p$, so that the expectation over all $b$ will be proportional to $4\epsilon \cdot \epsilon^p \sim \epsilon^{p+1}$. To ensure that the ratio between $\epsilon^{p+1}$ and $W_p(X_{2\epsilon}, X_\epsilon)^{\alpha p} \sim \epsilon^{\alpha p}$ will have a strictly positive lower bound as $\epsilon \to 0$, we need to choose $\alpha \geq \frac{p+1}{p}$. The proof that $\alpha = \frac{p+1}{p}$ is actually enough, and several other details necessary to turning this argument into a full proof, are given in the appendix.

**3.0.2 Adaptive ReLU** To attain the $(p + 1)/p$ exponent in theorem 3.1, we need to assume that the bias is drawn from an interval $[-B, B]$ which is large enough so that $\|x\| < B$ for all $x \in \Omega$. This assumption may be difficult to satisfy, especially in MPNNs where the features in intermediate layers are effected by previous parameter choices. Additionally, the $\pm\epsilon$ example shows that when $b$ is outside the range of the multiset features, $m_{\text{ReLU}}$ is not effective. Motivated by these observations, we propose a novel parametric function for multisets, based on $\text{ReLU}$, where the bias is automatically adapted to feature values. For a multiset $X = \{\!\{x_1, \ldots, x_r\}\!\}$ and $a \in \mathbb{R}^d, t \in [0, 1]$, the adaptive ReLU function $m_{\text{ReLU}}^{\text{adapt}}(X; a, t)$ is defined using $m = \min\{a \cdot x_1, \ldots, a \cdot x_r\}, M = \max\{a \cdot x_1, \ldots, a \cdot x_r\}, b = (1 - t)m + tM$ and

$$m_{\text{ReLU}}^{\text{adapt}}(X; a, t) = [r, m, M, \frac{1}{r}\sum_{i=1}^{r} \text{ReLU}(a \cdot x_i - b)]$$

The output of $m_{\text{ReLU}}^{\text{adapt}}$ is a four dimensional vector . The last coordinate of the vector is essentially $m_{\text{ReLU}}$, where the bias $b$ now only varies between the minimal and maximal value of multiset features. The first three coordinates are simple invariant features of the multisets: its cardinality and minimal and maximal value. In the appendix we prove

**Theorem 3.2.** *For $(n, d, p, \Omega, z)$ satisfying our standing assumptions, assume that $a \sim S^{d-1}$ and $t \sim [0, 1]$. Then the function $m_{\text{ReLU}}^{\text{adapt}} : \mathcal{S}_{\leq n}(\Omega) \times S^{d-1} \times [0, 1] \to \mathbb{R}^4$ is uniformly Lipschitz and $\frac{p+1}{p}$ lower-Hölder in expectation. Moreover, when $n \geq 4$, the function $m_{\text{ReLU}}^{\text{adapt}}$ is not $\alpha$ lower-Hölder in expectation for any $\alpha < \frac{p+1}{p}$.*

We note that while adaptive ReLU solves the $\pm\epsilon$ example, a similar issue can be created for adaptive ReLU with multisets such as $\{\!\{1, 0, 0, -1\}\!\}$ and $\{\!\{1, \epsilon, -\epsilon, -1\}\!\}$, which force the adaptive bias to cover the interval $[1, 1]$. Nonetheless, the delicate bias range assumption is no longer an issue.

**3.0.3 Summation with smooth activation** We now consider the Hölder properties of $m_\sigma$ when the activation $\sigma$ is smooth (e.g., sigmoid, SiLU, tanh). To understand this scenario, it can be instructive to first return to the $\pm\epsilon$ example. In this example, since the elements in $X_\epsilon$ and $X_{2\epsilon}$ both sum to the same number, one can deduce that (for any choice of bias) the first order Taylor expansion of $m_\sigma(X_{2\epsilon}) - m_\sigma(X_\epsilon)$ will vanish, so that this difference will be of the order of $\epsilon^2$. Based on this

example we can already see that for smooth activations $\alpha \geq 2$. However, it turns out that there are adversarial examples with much worse behavior. Specifically, note that $X_\epsilon$ and $X_{2\epsilon}$ are 'problematic' because their first moments are identical. In the same spirit, by choosing a pair $X, X'$ of distinct multisets with $n$ elements in $\mathbb{R}$, whose first $n-1$ moments are identical, we can deduce that $\alpha \geq n$:

**Theorem 3.3.** *Assume $(n, d, p, \Omega, z)$ satisfy our standing assumptions, $\sigma : \mathbb{R} \mapsto \mathbb{R}$ has $n$ continuous derivatives, and $a \sim S^{d-1}, b \sim [-B, B]$. If the function $m_\sigma(\cdot; a, b)$ is $\alpha$ lower-Hölder in expectation, then $\alpha \geq n$.*

**3.0.4 Sort based** We now present a fourth and final multiset parametric function. Based on ideas from (Balan et al., 2022; Dym & Gortler, 2023), we consider the parametric function

$$S_z(X; a, b) = b \cdot \text{sort}(a \cdot \rho_{(z)}(X)) \qquad (2)$$

where $X \in \mathbb{R}^{d \times n}$ and $a, b$ are in $S^{d-1}$ and $S^{n-1}$ respectively. Note that in contrast with our previous functions, $S_z$ explicitly includes the augmentation by $z$. On the bright side, the sort-based parametric functions are lower-Lipschitz in expectation.

**Theorem 3.4.** *For $(n, d, p, \Omega, z)$ satisfying our standing assumptions. Assume that $a \sim S^{d-1}$ and $b \sim S^{n-1}$. Then $S_z(\cdot; a, b)$ is uniformly upper Lipschitz and lower Lipschitz in expectation.*

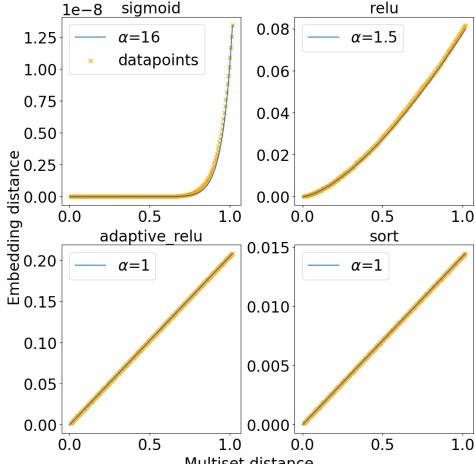

Figure 2: $l_2$ vs. $W_2$ distance on multiple adversarial multiset-pairs. Results are in accordance with our theoretical results (see Table1)

The proof of this claim is that such mappings were shown to be bi-Lipschitz for all parameters, when duplicated enough times (with independent parameters) (Balan et al., 2022). This implies bi-Lipschitzness in expectation for the duplicated function, which in turn implies bi-Lipschitzness in expectation for a single function since the relevant expectations are identical.

**Summary** In Figure 2, we choose a pair of distinct multisets $X, X'$ with 16 scalar features each, which have 15 identical moments (for details on how this was constructed see Subsection C.2 in the appendix). The figure plots the distance between embedding distances versus the Wasserstein distance, for pairs $\epsilon X, \epsilon X'$, for varying values of $\epsilon$. The figure illustrates that the $\alpha = 3/2$ Hölder exponent of ReLU (we take $p = 2$) and the $\alpha \geq n$ exponent of sigmoid summation are indeed encountered in this example. The sort and adaptive ReLU methods display a linear plot in this case. Recall that while sort is indeed lower-Lipschitz in expectation, adaptive ReLU is not. In Remark C.6 in the Appendix, we will explain why adaptive ReLU displays a linear plot in this case, and how a $3/2$ slope can be obtained by slightly changing this experiment.

## 4 MPNN Hölder Stability

We will now analyze the expected Hölder stability of MPNNs for graphs, using the various multiset embeddings discussed previously. We begin by defining MPNNs, WL tests, and a graph WL-metric.

### 4.1 MPNNs and WL

We denote a graph by $G = (V, E, X)$ where $V$ are the nodes, $E$ are the edges, and $X = (x_v)_{v \in V}$ are the initial node features. Throughout this section we will assume, as in the previous section, that $\Omega$ is a compact subset of $\mathbb{R}^d$, and $z$ is some fixed point in $\mathbb{R}^d \setminus \Omega$. We denote the neighborhood of a node $v$ by $\mathcal{N}_v = \{\!\!\{ u \in V | (u, v) \in E \}\!\!\}$. MPNNs iteratively use the graph structure to redefine node features in the following manner: initialize $x_v^{(0)} = x_v$.

**For k = 1 ..., K**

$$AGGREGATE: c_v^{(k)} = \phi^{(k)}(\{\!\!\{ x_u^{(k-1)} | u \in \mathcal{N}_v \}\!\!\})$$
$$COMBINE: x_v^{(k)} = \psi^{(k)}(x_v^{(k-1)}, c_v^{(k)})$$

In order to achieve a final graph embedding, a final *READOUT* step is performed

$$READOUT : c^{\text{global}} = \eta(\{\!\{x_v^{(K)}|v \in V\}\!\})$$

We note that different instantiations of the COMBINE, AGGREGATE, and READOUT parametric functions will lead to different architectures. We will discuss several choices, and their expected Hölder stability properties, slightly later on. Once such an instantiation is chosen, an MPNN will be a parametric model of the form $f(G; w) = c^{\text{global}}$, where $w$ denotes the concatenation of all parameters of the COMBINE, AGGREGATE, and READOUT functions.

The *Weisfeiler-Lehman* (WL) graph isomorphism test (Weisfeiler & Lehman, 1968; Huang & Villar, 2021) checks whether two graphs $G, H$ are isomorphic (identical up to node permutation) by applying an MPNN-like procedure while choosing the COMBINE, AGGREGATE and READOUT functions to be hash functions, and running the procedure for at most $K = max\{|V_G|, |V_H|\}$ (Morris et al., 2019) iterations. The WL test can separate many, but not all, pairs of non-isomorphic graphs.

Notably, MPNNs can only separate pairs of graphs which are WL separable, and therefore if we hope for MPNN Hölder stability in expectation, we need to choose a *WL metric*: a pseudo-metric on graphs which is zero if and only if the pair of graphs cannot be separated by WL. Of the several possible choices in the recent literature for such a metric (Grohe, 2020; Böker, 2021; Chen et al., 2022), we choose the family of Tree Mover's Distance ($TMD^{(K)}$) suggested in (Chuang & Jegelka, 2022), which is described in detail in Appendix E. $TMD^{(K)}$ is a distance which is non-zero if and only if two graphs can be separated by $K$ iterations of the WL test. It is based on recursive application of the augmented Wasserstein to multisets of WL computation trees. We note that when $K$ is large enough, $TMD^{(K)}$ is a WL metric as discussed above.

## 4.2 MPNN STABILITY ANALYSIS

We now consider the Hölder stability of MPNNs. The graph domain we consider are graphs in $G_{\leq n}(\Omega)$, the collection of graphs with up to $n$ nodes, and node features in a compact set $\Omega \subseteq \mathbb{R}^d$.

It is natural to expect that the stability properties of an MPNN will be closely related to the AGGREGATE, READOUT and COMBINE functions composing the MPNN. As a COMBINE function, we will choose some parametric functions which is uniformly upper-Lipschtiz, and lower Lipschitz in expectation. As the COMBINE function is a vector-to-vector function, this is easy to achieve even via a linear function $x_v^{(k)} = Ax_v^{(k-1)} + Ba_v^{(k)}$. In Appendix D we prove the Lipschitz properties of this and three other COMBINE choices.

The AGGREGATE and READOUT functions are multiset to vector functions, and we have discussed the Hölder properties of four different such choices in Section 3. In practice, at each iteration $k$ we take $W_k$ parallel copies of these functions with independent parameters ($W$ corresponds to the width of the MPNN). As discussed in the end of Subsection 2.1, this maintains the same results in expectation while reducing the variance. For example, the SortMPNN architecture will be defined using the sort-based parametric functions $S_z$ from (2) via

$$AGGREGATE: c_{v,i}^{(k)} = S_{z_k}\left(\{\!\{x_u^{(k-1)}|u \in \mathcal{N}_v\}\!\}; a_i^{(k)}, b_i^{(k)}\right), \quad i = 1, \ldots W_k$$

$$COMBINE: x_v^{(k)} = A^{(k)}x_v^{(k-1)} + B^{(k)}c_v^{(k)}$$

$$READOUT : c_i^{\text{global}} = S_{z_{K+1}}\left(\{\!\{x_v^{(K)}|v \in V\}\!\}; a_i^{(K+1)}, b_i^{(K+1)}\right), \quad i = 1, \ldots, W_{K+1}$$

Similarly, we will use the terms AdaptMPNN, ReluMPNN and SmoothMPNN to denote the MPNN obtained by replacing $S_z$ with the appropriate functions $m_{\text{ReLU}}^{\text{adapt}}$, $m_{\text{ReLU}}$ or $m_\sigma$ with a smooth $\sigma$.

Our first result regards the upper Lipschitz bound.

**Theorem 4.1.** *(Uniformly Lipschitz MPNN embeddings, informal version) Let $f : G_{\leq n}(\Omega) \to \mathbb{R}^m$ be an MPNN with $K$ layers. If the functions used for the aggregation $\phi^{(k)}$, combine $\psi^{(k)}$, and readout $\eta$ are all uniformly upper Lipschitz, then $f$ is uniformly upper Lipschitz with respect to $TMD^{(K)}$. In particular, ReluMPNN, SmoothMPNN, AdaptMPNN and SortMPNN are all uniformly upper Lipschitz.*

It would seem natural to expect that similar results will hold for lower-Hölder in expectation guarantees: namely, that if the AGGREGATE, COMBINE and READOUT functions used in the MPNN are lower-Hölder in expectation, then the overall MPNN will be lower-Hölder in expectation as well. [2] Unfortunately, this isn't always the case. In Appendix F.2 we give an example of a pair of graphs which cannot be separated by any choice of parameter of a Relu-MPNN with width 1 and depth 2, even though they are separated by 2 iterations of WL.

The argument above does not rule out the possibility that a wider Relu-MPNN will be lower-Hölder in expectation. Indeed, we suspect that this will be the case for an appropriate width $W = W(d, n)$, but we leave a formal proof of this result for future work. We do prove that, for arbitrarily wide ReluMPNN and SmoothMPNN, we cannot obtain a very good Hölder exponent, with a bound that deteriorates as the depth increases:

**Theorem 4.2.** *Assume that ReluMPNN with depth $K$ is $\alpha$ lower-Hölder in expectation with respect to $TMD^{(K)}$, then $\alpha \geq 1 + \frac{K+1}{p}$. If SmoothMPNN with depth $K$ is $\alpha$ lower-Hölder in expectation then $\alpha \geq 2^{K+1}$.*

The proof of this theorem is based on an adversarial set of examples we call $\epsilon$-Trees. These are defined recursively, where the first set of trees are of height two, and the leaves contain the $\pm\epsilon$ multisets. Deeper trees are then constructed by building upon substructures from the trees in the previous step as depicted in figure 6 alongside a rigorous formulation and proof in Appendix F.3.

In contrast to the previously discussed methods, SortMPNN is lower Lipschitz in expectation, even with a width of 1.

**Theorem 4.3.** *(informal) For any given $W \geq 1, K > 0$, SortMPNN with width $W$ and depth $K$ is lower Lipschitz in expectation with respect to $TMD^{(K)}$.*

While we don't formally analyze the lower-Hölder properties of AdaptMPNN, we conjecture its worst-case behavior will be similar to ReluMPNN. However, in some settings it will have better stability. For example, for our adversarial $\epsilon$-Trees example AdaptMPNN has a Lipschitz-like behavior, as shown in Figure 3.

## 5 EXPERIMENTS

In the following experiments, we evaluate SortMPNN, AdaptMPNN, ReluMPNN and SmoothMPNN. As the last two architectures closely resemble standard MPNN like GIN (Xu et al., 2019) with ReLU/smooth activation, our focus is mainly on the SortMPNN and AdaptMPNN architectures. In our experiments we consider several variations of these architectures which were omitted in the main text for brevity, and are described in appendix G alongside further experiment details.

$\epsilon$**-Tree dataset** In order to show that the expected (lower) Hölder exponent is indeed a good indicator of separation quality, and to further validate the importance of separation quality analysis, we first focus on the adversarial $\epsilon$-tree construction used to prove Theorem 4.2.

To begin with, we show how randomly initialized MPNNs with the different multiset embeddings we analyzed distort the TMD metric on the $\epsilon$-trees. In figure 3 we plot the distance between the embeddings provided by the MPNNs, as a function of the parameter $\epsilon$ determining the $\epsilon$ trees (this parameter is proportional

Table 2: $\epsilon$-Tree binary classification results

| Model | Accuracy |
|---|---|
| GIN(Xu et al., 2019) | 0.5 |
| GCN(Kipf & Welling, 2017) | 0.5 |
| GAT(Velickovic et al., 2018) | 0.5 |
| ReluMPNN | 0.5 |
| SmoothMPNN | 0.5 |
| SortMPNN | 1.0 |
| AdaptMPNN | 1.0 |

to the TMD distance between the trees). We see how the lower-Hölder exponent increases with depth for ReluMPNN and SmoothMPNN, in a manner which is consistent with the $\alpha = 1 + \frac{K+1}{p}$ and $2^{K+1}$ bounds suggested by Theorem 4.2 (for $p = 2$). SortMPNN, in contrast, displays virtually no distortion in this example, and its behavior is consisten with the bi-Lipschitness predicted by our theory. Similar results (without theoretical justification) can be observed for AdaptMPNN.

We then proceed to train a binary classifier using these MPNNs in an attempt to separate the tree pairs.

---

[2] Indeed, we made this claim in a previous version. We thank Soutrik Sarangi and Abir De for pointing out the error in the proof.

Table 3: Classification accuracy on TUdatasets (Morris et al., 2020). Best in **bold**, second underlined.

| Dataset | Mutag | Proteins | PTC | NCI1 | NCI109 |
|---|---|---|---|---|---|
| GIN(Xu et al., 2019) | 89.4±5.6 | 76.2±2.8 | 64.6±7 | 82.7±1.7 | 82.2±1.6 |
| GCN(Kipf & Welling, 2017) | 85.6±5.8 | 76±3.2 | 64.2±4.3 | 80.2±2.0 | NA |
| GraphSage(Hamilton et al., 2017) | 85.1±7.6 | 75.9±3.2 | 63.9±7.7 | 77.7±1.5 | NA |
| SortMPNN | **90.99±6.2** | **76.46±3.68** | 66.31±6.73 | **83.55±1.82** | 82.75±1.60 |
| AdaptMPNN | 90.41±6.1 | 75.12±3.64 | **66.87±5.37** | 82.77±1.72 | **83.26±0.86** |

The results in Table 2 show how SortMPNN and AdaptMPNN achieve perfect performance, while ReluMPNN and SmoothMPNN, in addition to several baseline MPNNs completely fail. This serves as proof to the importance of separation quality analysis, especially in light of the theoretical capability of SmoothMPNN (Amir et al., 2023) and GIN (Xu et al., 2019) to separate these graphs.

**TUDataset** While the $\epsilon$-Tree dataset emphasizes the importance of high separation quality, validating our analysis, it is not obvious what effect this has in real world datasets. Therefore, we test SortMPNN and

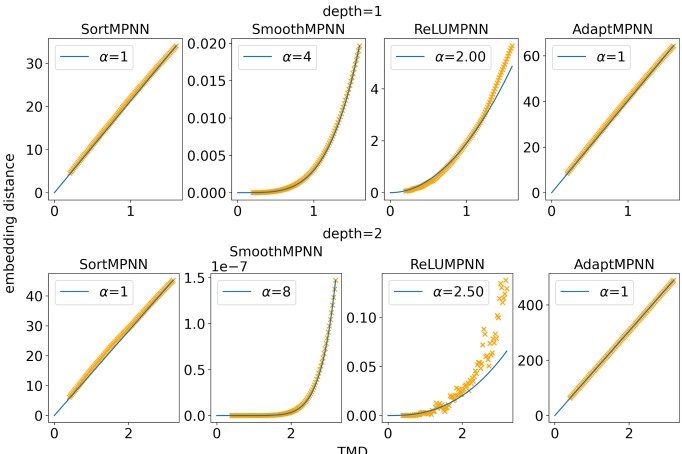

Figure 3: $l_2$ vs. TMD on $\epsilon$-Trees. The targeted ReluMPNN and SmoothMPNN exponents deteriorate with depth, in accordance with our theory (see Table 1)

AdaptMPNN on a subset of the TUDatasets (Morris et al., 2020), including Mutag, Proteins, PTC, NCI1 and NCI109. Table 3 shows that SortMPNN and AdaptMPNN outperform several baseline MPNNs (results are reported using the evaluation method from (Xu et al., 2019)). Note that in all experiments we only compare to Vanilla MPNNs (1-WL), and not more expressive architectures which often do reach better results, at a higher computational cost.

**LRGB** We further evaluate our architectures on two tasks from LRGB (Dwivedi et al., 2022b): peptides-func, which is a multi-label graph classification task, and the peptides-struct regression task. Results in table 4 show that SortMPNN and AdaptMPNN outperform other methods on peptides-func. For peptides-struct, GCN is the best performing method with SortMPNN a close second.

Both experiments above adhere to the 500K parameter constraint as in (Dwivedi et al., 2022b). In addition, we reran the peptides-struct experiment with a 100K, 50K, 25K, 7K and 1K parameter budget. The results in figure 8 in the appendix show SortMPNN and AdaptMPNN outperforming GCN for smaller models, with SortMPNN achieving the best results. We believe this is related to the fact that even a SortMPNN with width=1 is lower-Lipschitz in expectation, and accordingly when only a small number of features is available, its advantage on other methods is more substantial.

Table 4: LRGB results. Best in **bold**, Second underlined.

| Dataset | peptides-func (AP↑) | peptides-struct (MAE↓) |
|---|---|---|
| GINE(Hu* et al., 2020) | 0.6621±0.0067 | 0.2473±0.0017 |
| GCN(Kipf & Welling, 2017) | 0.6860±0.0050 | **0.2460±0.0007** |
| GatedGCN(Bresson & Laurent, 2018) | 0.6765±0.0047 | 0.2477±0.0009 |
| SortMPNN | **0.6940±0.0049** | 0.2464±0.0024 |
| AdaptMPNN | 0.6934±0.0099 | 0.2484±0.0034 |

**Subgraph aggregation networks - Zinc12K**  Finally, we experiment the use of our methods as the backbone MPNN in a more advanced equivariant subgraph aggregation network (ESAN) (Bevilacqua et al., 2022), which has greater separation power than MPNNs.  To this extent, we run the experiment from (Bevilacqua et al., 2022) on the ZINC12K dataset, where we swap the base encoder from GIN (Xu et al., 2019) to SortMPNN and AdaptMPNN. The results shown in table 5 show that SortMPNN outperforms GIN in five of the eight different scenarios.  For fair comparison, the models don't surpass the 100K parameter budget.

**Code and timing**  Timing for the LRGB experiment are provided in table 7 in the appendix.  SortMPNN is marginally slower than GCN, and AdaptMPNN is X1.87 times slower. Code is available at [3] .

Table 5: MAE per base encoder for ESAN on ZINC12K. Best in **bold**, second underlined.

| Method | GIN (Xu et al., 2019) | SortMPNN | AdaptMPNN |
|---|---|---|---|
| DS-GNN (ED) | 0.172±0.008 | **0.157±0.007** | 0.176±0.008 |
| DS-GNN (ND) | 0.171±0.010 | **0.152±0.009** | 0.168±0.008 |
| DS-GNN (EGO) | 0.126±0.006 | **0.104±0.004** | 0.127±0.007 |
| DS-GNN (EGO+) | 0.116±0.009 | **0.115±0.008** | 0.126±0.007 |
| DSS-GNN (ED) | 0.172±0.005 | **0.169±0.004** | 0.173±0.007 |
| DSS-GNN (ND) | **0.166±0.004** | 0.167±0.006 | 0.167±0.008 |
| DSS-GNN (EGO) | **0.107±0.005** | 0.115±0.010 | 0.126±0.007 |
| DSS-GNN (EGO+) | **0.102±0.003** | 0.121±0.005 | 0.131±0.006 |

## 6  RELATED WORK

**Sorting**  Sorting based operations were used for permutation invariant networks on multisets in (Zhang et al., 2019), and as readout functions for MPNNs in (Balan et al., 2022; Zhang et al., 2018; Duvenaud et al., 2015).  To the best of our knowledge, our SortMPNN is the first network using sorting for aggregations, and the first MPNN with provable bi-Lipschitz (in expectation) guarantees.

**Bi-Lipschitz stability**  An alternative bi-Lipschitz embedding for multisets was suggested in (Cahill et al., 2024a).  This embedding has higher computational complexity then sort based embedding. Some additional examples of recent works on bi-Lipschitz embeddings include (Balan & Tsoukanis, 2023; Cahill et al., 2024a; 2020). (Upper) Lipschitz stability of graph neural networks from a spectral perspective is discussed in Gama et al. (2020); Pfrommer et al. (2021).

We note that (Böker et al., 2023) does provide lower and upper stability estimates for MPNNs with respect to a WL metric. These stability estimates are in an $\epsilon - \delta$ sense, which do not rule out arbitrarily bad Hölder exponents. Moreover, they only consider graphs without node features. On the other hand, their analysis is more general in that they consider graphs of arbitrary size, and their graphon limit. In an additional recent work, Xu et al. (2023) introduced a GNN designed to be bi-Lipschitz with respect to a weighted inner product space. Their approach, however, is limited to scenarios with a fixed graph topology, where only scalar node features vary. This fixed topology justifies the use of the weighted inner product space, which does not incorporate information about the graph structure.

## 7  SUMMARY AND LIMITATIONS

We presented expected Hölder stability analysis for functions based on ReLU summation, smooth activation summation, adaptive ReLU, and sorting. Our theoretical and empirical results suggest SortMPNN as a promising alternative to traditional sum-based MPNNs.

A computational limitation of SortMPNN is that it requires prior knowledge of maximal multiset sizes for augmentation. We believe that future work will reveal ways of achieving lower-Lipschitz architectures without augmentation. Other avenues of future work include analyzing Hölder properties with respect to other graph metrics other than TMD, and resolving some questions we left open such as upper bounds for the Hölder exponents of smooth activations.

**Acknowledgements:** N.D. and Y.D. are supported by Israeli Science Foundation grant no. 272/23. We thank Haggai Maron and Guy Bar-Shalom for their useful insights and assistance.

---

[3]https://github.com/YDavidson/On-The-Holder-Stability

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

## A   DEFINITIONS AND NOTATION

The following definitions and notation are used throughout the appendices.

- We use the function $F(x, x', w) = F_{p,\alpha,f}(x, x', w)$ to denote

$$F(x, x', w) = \left\{ \frac{d_Y(f(x, w), f(x', w))}{d_X(x, x')^\alpha} \right\}^p$$

- We call a pair of multisets *balanced*, if the number of elements they contain are equal, and otherwise we call them *unbalanced*.

## B   HÖLDER STABILITY IN EXPECTATION PROPERTIES

In this section we fill in the details of some properties of lower-Hölder in expectation functions, which were discussed in Subsection 2.1.

### B.1   REDUCING VARIANCE BY AVERAGING

Given $N \in \mathbb{N}$, we can extend $f(x; w) : X \times W \to Y$ to a new parametric function $f_N : X \times W^N \to Y^N$ defined as

$$f_N(x; w_1, \ldots, w_N) = [f(x, w_1), \ldots, f(x, w_N)] \tag{3}$$

where the f measure on $W^N$ is the product measure, and the distance we take on $Y^N$ is

$$d_{Y_N}([y_1, \ldots, y_N], [y'_1, \ldots, y'_N]) = \left\{ \frac{1}{N} \sum_{n=1}^N d(y_n, y'_n)^p \right\}^{1/p}$$

The function $F_N$ corresponding to this choice of $f_N$ is

$$F_N(x, x', w) = \left\{ \frac{d_{Y_N}(f_N(x, \mathbf{w}), f_N(x', \mathbf{w}))}{d_X(x, x')^\alpha} \right\}^p$$

$$= \left( \frac{1}{d_X(x, x')} \right)^{\alpha p} \frac{1}{N} \sum_{n=1}^N d(f(x, w_n), f(x', w_n))^p$$

$$= \frac{1}{N} \sum_{n=1}^N F(x, x', w_n)$$

Thus $F_N$ is the average of $N$ independent copies of $F$, which implies that for all $x \neq x'$ the random variable $F_N(x, x', \mathbf{w})$ has the same expectation as $F$, and its standard deviation is uniformly bounded

by $\sigma/\sqrt{N}$, where $\sigma$ is a bound on the standard deviation of $F(x, x', w)$ which is uniform in $x, x'$ (assuming that such a bound exists). In particular, this means that *for any* $x, x'$, not only will the expectation of $F_N(x, x', \mathbf{w})$ be bounded by from below by $m^p$, but also as $N$ goes to infinity the probability of $F_N(x, x', \mathbf{w})$ being larger than $m^p - \epsilon$ will go to one.

## B.2 BOUNDEDNESS ASSUMPTIONS

In the scenarios we consider in this paper the metric spaces are bounded. Assume $0 < \alpha < \beta$, and the distances between any two elements $x, x'$ in a metric space $X$ are bounded by some constant $B$, then

$$d(x, x')^\beta = B^\beta \left( \frac{d(x, x')}{B} \right)^\beta \leq B^\beta \left( \frac{d(x, x')}{B} \right)^\alpha = B^{\beta - \alpha} d(x, x')^\alpha.$$

In other words, in bounded metric spaces, up to a constant, increasing the exponent decreases the value. It follows that a parametric functions which is $\alpha$ lower Hölder in expectation is also $\beta$ lower Hölder in expectation.

## B.3 COMPOSITION

In this section we discuss what happens when we compose two functions $f(x; w) : X \times W \mapsto Y$ and $g(y; v) : Y \times V \mapsto Z$ to get a new parametric function

$$g \circ f(x; w, v) = g(f(x; w); v).$$

**Lemma B.1.** *Given two parametric functions* $f(x; w) : X \times W \mapsto Y$ *and* $g(y; v) : Y \times V \mapsto Z$ *which are* $\alpha$ *and* $\beta$ *lower Hölder in expectation with* $\alpha, \beta \geq 1$, *then* $g \circ f$ *is* $\alpha \cdot \beta$ *lower Hölder in expectation. Similarly, if* $f, g$ *are uniformly upper Lipschitz, then* $f \circ g$ *is uniformly upper Lipschitz.*

*Proof.* We omit the proof for uniform upper Lipschitz, and prove only the more challenging case of lower-Hölder in expectation. We have that for strictly positive $m_f, m_g$, $\mathbb{E}_{w \sim \mu}[\{\frac{d_Y(f(x;w),f(x';w))}{d_X(x,x')^\alpha}\}^p] \geq m_f$ and $\mathbb{E}_{v \sim \gamma}[\{\frac{d_Z(g(y;v),g(y';v))}{d_Y(y,y')^\beta}\}^p] \geq m_g$, from which we conclude

$$\mathbb{E}_{w,v \sim (\mu,\gamma)}[\{\frac{d_Z(g(f(x;w),v),g(f(x';w),v))}{d_X(x,x')^{\alpha \cdot \beta}}\}^p]$$

$$\overset{\text{w,v are independent}}{=} \int_w \int_v \{\frac{d_Z(g(f(x;w),v),g(f(x';w),v))}{d_X(x,x')^{\alpha \cdot \beta}}\}^p d\gamma d\mu$$

$$= \int_w \{\frac{d_Y(f(x;w),f(x';w))^\beta}{d_X(x,x')^{\alpha \cdot \beta}}\}^p \int_v \{\frac{d_Z(g(f(x;w),v),g(f(x';w),v))}{d_Y(f(x;w),f(x';w))^\beta}\}^p d\gamma d\mu$$

$$= \int_w \{\frac{d_Y(f(x;w),f(x';w))^\beta}{d_X(x,x')^{\alpha \cdot \beta}}\}^p \cdot \mathbb{E}_{v \sim \gamma}[\{\frac{d_Z(g(f(x;w),v),g(f(x';w),v))}{d_Y(f(x;w),f(x';w))^\beta}\}^p] d\mu$$

$$\geq \int_w \{\frac{d_Y(f(x;w),f(x';w))^\beta}{d_X(x,x')^{\alpha \cdot \beta}}\}^p \cdot m_g d\mu$$

$$= m_g \cdot \mathbb{E}_{w \sim \mu}[(\{\frac{d_Y(f(x;w),f(x';w))}{d_X(x,x')^\alpha}\}^p)^\beta]$$

$$\overset{(*)}{\geq} m_g \cdot (\mathbb{E}_{w \sim \mu}[\{\frac{d_Y(f(x;w),f(x';w))}{d_X(x,x')^\alpha}\}^p])^\beta$$

$$\geq m_g \cdot m_f^\beta$$

Where $(*)$ is true by Jensen's inequality, since $(\frac{d_Y(f(x;w),f(x';w))}{d_X(x,x')^\alpha})^p$ is non negative, and $\phi : \mathbb{R}^+ \cup \{0\} \mapsto \mathbb{R}, \phi(t) = t^\beta$ is convex for $\beta \geq 1$

$\square$

## C  MULTISET EMBEDDINGS ANALYSIS PROOFS

The following are the proofs of the claims regarding the exponent of Hölder stability in expectation from section 3.

Before stating and proving the relevant claims, we will present some lemmas that will be used throughout this section and those that follow.

We first state the well known property of norm equivalence on finite dimensional normed spaces

**Lemma C.1.** *Let $1 \leq p, q \leq \infty$ and $d \in \mathbb{N}$. There exists $c_{p,q,d} > 0$ s.t. for any $v \in \mathbb{R}^d$*

$$\|v\|_p \geq c_{p,q,d} \cdot \|v\|_q$$

In particular, this means that

**Lemma C.2.** *Let $1 \leq p, q \leq \infty$ and $\Omega \subseteq \mathbb{R}^d$ There exists $C_{p,q,d} > 0$ s.t. for any $X, Y \in \mathcal{S}_{\leq n}(\Omega)$*

$$W_p^{(z)}(X, Y) \geq C_{p,q,d} \cdot W_q^{(z)}(X, Y)$$

*Proof.*

$$
W_p^{(z)}(X, Y) = \left\{ \min_{\tau \in S_n} \sum_{j=1}^n \|\rho_{(z)}(x)_j - \rho_{(z)}(y)_{\tau(j)}\|_p^p \right\}^{1/p}
$$

$$
\overset{C.1}{\geq} c_{p,1,d} \cdot \left\{ \min_{\tau \in S_n} \sum_{j=1}^n \|\rho_{(z)}(x)_j - \rho_{(z)}(y)_{\tau(j)}\|_p \right\}
$$

$$
\overset{C.1}{\geq} c_{p,1,d} \cdot c_{p,q,d} \cdot \left\{ \min_{\tau \in S_n} \sum_{j=1}^n \|\rho_{(z)}(x)_j - \rho_{(z)}(y)_{\tau(j)}\|_q \right\}
$$

$$
\overset{C.1}{\geq} c_{p,1,d} \cdot c_{p,q,d} \cdot c_{q,1,d} \cdot \left\{ \min_{\tau \in S_n} \sum_{j=1}^n \|\rho_{(z)}(x)_j - \rho_{(z)}(y)_{\tau(j)}\|_q^q \right\}^{\frac{1}{q}}
$$

$$
= c_{p,1,d} \cdot c_{p,q,d} \cdot c_{q,1,d} \cdot W_q^{(z)}(X, Y)
$$

The upper bound can be proven in the same manner. □

Now that we are equipped with the above lemmas, we begin by proving for smooth functions.

### C.1  SUMMING OVER A SMOOTH ACTIVATION

**Theorem 3.3.** *Assume $(n, d, p, \Omega, z)$ satisfy our standing assumptions, $\sigma : \mathbb{R} \mapsto \mathbb{R}$ has $n$ continuous derivatives, and $a \sim S^{d-1}, b \sim [-B, B]$. If the function $m_\sigma(\cdot; a, b)$ is $\alpha$ lower-Hölder in expectation, then $\alpha \geq n$.*

*Proof.* It is sufficient to prove the claim in the case where $d = 1$.

Since $\sigma$ is $n$ times continuously differentiable, there exists some constant $C$ such that

$$|\frac{\partial^n}{\partial t^n}\sigma(at + b)| \leq C, \forall(t, a, b) \in [-1, 1] \times S^{d-1} \times [-B, B].$$

Now, let $f$ be any function with $n$ continuous derivatives and $|f^{(n)}(t)| < C$ for all $t \in [-1, 1]$. For $x \in (-1, 1)^n$ denote $F(x) = \sum_{i=1}^n f(x_i)$. (when $f(t) = \sigma(at + b)$ we have $F(x) = m_\sigma(x; a, b)$). By Taylor's approximation of $f$ around 0, there exists points $c_1, \ldots, c_n \in (-1, 1)$ such that

$$
F(x) = \sum_{i=1}^n \left( f(0) + f'(0)x_i + \frac{f''(0)}{2}x_i^2 + \ldots + \frac{f^{(n-1)}(0)}{(n-1)!}x_i^{n-1} + \frac{f^{(n)}(c_i)}{n!}x_i^n \right)
$$

$$
= nf(0) + f'(0)\sum_{i=1}^n x_i + \frac{f''(0)}{2}\sum_{i=1}^n x_i^2 + \ldots + \frac{f^{(n-1)}(0)}{(n-1)!}\sum_{i=1}^n x_i^{n-1} + \sum_{i=1}^n x_i^n \frac{f^{(n)}(c_i)}{n!}
$$

It follows that if $x, y$ are two vectors which are not the same, even up to permutation, but their first $n - 1$ moments are identical, then for every $\epsilon \in (0, 1)$, the vectors $\epsilon x$ and $\epsilon y$ will also have the same $n - 1$ moments, and so there exist $c_1, \ldots, c_n, d_1, \ldots, d_n$ in $(-1, 1)$ such that

$$|F(\epsilon x) - F(\epsilon y)| = \epsilon^n |\sum_{i=1}^n \frac{f^{(n)}(c_i)}{n!} x_i^n + \sum_{i=1}^n \frac{f^{(n)}(d_i)}{n!} y_i^n| \leq \frac{2n}{n!} C \epsilon^n$$

In contrast, the Wasserstein distance between $\epsilon x$ and $\epsilon y$ scales like $\epsilon$:

$$W_p^z(\epsilon x, \epsilon y) = W_p(\epsilon x, \epsilon y) = \epsilon W_p(x, y)$$

It follows that for all $\alpha < n$, all parameters $w = (a, b) \in S^{d-1} \times [-B, B]$, and all $\epsilon \in (0, 1)$,

$$\left| \frac{m_\sigma(\epsilon x; w) - m_\sigma(\epsilon y; w)}{W_p^\alpha(\epsilon x, \epsilon y)} \right|^p \leq C W_p(x, y)^{-\alpha} \frac{2n}{n!} \epsilon^{(n-\alpha)p}$$

Taking the expectation over the parameters $w$ on the left hand side, we get the same inequality. Taking the limit $\epsilon \to 0$, we obtain

$$\mathbb{E}_w \left| \frac{m_\sigma(\epsilon x; w) - m_\sigma(\epsilon y; w)}{W_p^\alpha(\epsilon x, \epsilon y)} \right|^p \leq C W_p(x, y)^{-\alpha} \frac{2n}{n!} \epsilon^{(n-\alpha)p} \to 0$$

from which we deduce that $m_\sigma$ is not $\alpha$ lower-Hölder in expectation.

To conclude this argument , we note that there always exist $x, y \in \mathbb{R}^n$ which are not identical up to permutation, but have the same first $n - 1$ moments. This is because it is known that no mapping from $(-1, 1)^n$ to $\mathbb{R}^n$, and in particular the mapping which takes a vector of length $n$ to its first $n - 1$ moments, cannot be injective, up to permutations (Wagstaff et al., 2019). To ensure that $x, y$ are in $(-1, 1)^n$, we can scale both vectors by a sufficiently small positive number. In the next subsection we discuss a constructive method for producing pairs of sets with $n - 1$ equal moments. $\qquad \square$

## C.2 Creating sets with $n - 1$ equal moments

In the proof of theorem 3.3 we explained that for every natural $n$, there exists pairs of vectors $x, y \in \mathbb{R}^n$ which are not identical, up to permutation, but have the same first $n - 1$ moments. We now give a constructive algorithm to construct such pairs for $n$ which is a power of 2, that is, $n = 2^k$ for some natural $k$. This algorithm was used to produce figure 2 in the main text.

Our algorithm operates by recursion on $k$. For $k = 1$, our goal is to find a pair of vectors $x^{(1)}, y^{(1)}$ of length $2^k = 2$, whose first $2^k - 1 = 1$ moments are equal. We can choose for example

$$x^{(1)} = [-1, 1], y^{(1)} = [-2, 2].$$

Now, assume that for a given $k$ we have a pair of vectors $x = x^{(k)}$ and $y = y^{(k)}$ of length $n = 2^k$ which are not identical, up to permutation, but whose first $2^k - 1$ moments are identical. We then perform the following three steps:

**step 1:** We translate all coordinates of $x$ and $y$ by the same number $t$, which we take to be the minimum of all coordinates in $x$ and $y$. We thus get a new pair of vectors

$$x' = [x_1 - t, \ldots, x_n - t], \quad y' = [x_1 - t, \ldots, x_n - t]$$

whose coordinates are all non-negative. These vectors are still distinct, even up to permutation, and have the same first $n - 1 = 2^k - 1$ moments.

**step 2:** We take the root of the non-negative entries in both vectors, to get new vectors

$$x'' = [\sqrt{x_1'}, \ldots, \sqrt{x_n'}], \quad y'' = [\sqrt{y_1'}, \ldots, \sqrt{y_n'}].$$

These two vectors are distinct, even up to permutations, and they agree on the even moments $2, 4, \ldots, 2(n-1)$.

**step 3:** Finally, we define a new pair of vectors of cardinality $2n = 2^{k+1}$

$$x^{(k+1)} = [-x'', x''], \quad y^{(k+1)} = [-y'', y'']$$

These two new vectors are still distinct up to permutations. There even moments up to order $2(n-1)$ are still identical. Moreover, all odd moments of both vectors are zero, and hence identical. In particular, the two vectors have the same moments of order $1, 2, \ldots, 2(n-1), 2n - 1$, which is what we wanted.

## C.3   SORTING

Our goal in this subsection is proving

**Theorem 3.4.** *For $(n, d, p, \Omega, z)$ satisfying our standing assumptions. Assume that $a \sim S^{d-1}$ and $b \sim S^{n-1}$. Then $S_z(\cdot; a, b)$ is uniformly upper Lipschitz and lower Lipschitz in expectation.*

Recall that $S_z$ is defined via

$$S_z(X; a, b) = b \cdot \text{sort}(a \cdot \rho_{(z)}(X))$$

Once $\rho_{(z)}$ is applied, we can think of $S_z$ as operating on multisets with exactly $n$ elements, coming from the domain $\Omega \cup \{z\}$. The function $S_z$ is then a composition of parametric functions $L \circ s$, where

$$L(y; b) = b \cdot y \tag{4}$$

and

$$s(X; a) = sort(a^T X) \tag{5}$$

By the rules of composition (Lemma B.1), it is sufficient to show that both functions are uniformly Lipschitz, and lower-Lipschitz in expectation.

We first show for $L$

**Lemma C.3.** *For every $p \geq 1$, the function $L : \mathbb{R}^n \times S^{n-1} \to \mathbb{R}$ is uniformly upper-Lipschitz, and lower Lipschitz in expectation.*

*Proof.* **Uniformly upper Lipschitz** For every $b \in S^{n-1}$ and $x, x' \in \mathbb{R}^n$ we have, using Cauchy-Schwarz,

$$|L(x; b) - L(x'; b)| = |b \cdot (x - x'),| \leq \|x - x'\|$$

so for all $b$ we have a Lipschitz constant of 1.

**Lower Lipschitz in expectation** For every $x \neq x'$ in $\mathbb{R}^n$ we have, due to lemma C.1, for the appropriate $c > 0$

$$\mathbb{E}_{b \sim S^{n-1}} \left[ \left\{ \frac{|L(x; b) - L(x'; b)|}{\|x - x'\|_p} \right\}^p \right] \geq c^p \cdot \mathbb{E}_{b \sim S^{n-1}} \left[ \left\{ \frac{|L(x; b) - L(x'; b)|}{\|x - x'\|_2} \right\}^p \right]$$

$$= c^p \cdot \mathbb{E}_{b \sim S^{n-1}} \left[ |b \cdot \frac{x - x'}{\|x - x'\|_2}|^p \right]$$

$$= c^p \cdot \mathbb{E}_{b \sim S^{n-1}} \left[ |b \cdot e_1|^p \right] > 0$$

where the last equality is because the distribution is rotational invariant. $\qquad\square$

We now prove for $s$

**Lemma C.4.** *Let $d, n, p$ be natural numbers. Let $s$ be as defined in (5), then $s$ is uniformly upper Lipschitz, and lower Lipschitz in expectation. (here, the domain of $s$ is the space $\mathcal{S}_{=n}(\Omega)$ of multisets with $n$ elements in a compact set $\Omega \subseteq \mathbb{R}^d$, endowed with the $W_1$ metric. $a$ is drawn uniformly from $S^{d-1}$, and the metric on the output of $s$ is the $\ell_p$ distance).*

*Proof.* Due to norm equivalence as stated in lemma C.1, it is sufficient to prove the claim when $p = 2$.

Fix some balanced multisets $X \nsim Y$. Let $\sigma$ be a permutation which minimizes $\sum_{j=1}^n \|x_j - y_{\tau(j)}\|_2$ over all $\tau \in S_n$. Then for every $a \in S^{d-1}$ we have, using Cauchy-Schwartz and norm equivalence

C.1

$$\|s(X;a) - s(Y;a)\|_2^2 = \|sort(a^T X) - sort(a^T Y)\|_2^2$$

$$= \min_{\tau \in S_n} \sum_{j=1}^{n} |a^T x_j - a^T y_{\tau(j)}|^2$$

$$\leq \sum_{j=1}^{n} |a^T x_j - a^T y_{\sigma(j)}|^2$$

$$\leq \sum_{j=1}^{n} \|x_j - y_{\sigma(j)}\|_2^2$$

$$\leq C \left( \sum_{j=1}^{n} \|x_j - y_{\sigma(j)}\|_2 \right)^2$$

$$= W_1(X,Y)^2$$

Concluding $s$ is uniformly upper Lipschitz.

In addition, we also have that for large enough $N$, the function $s_N(\cdot;a)$ (concatenation of $s$ $N$ times, divided by $1/N$) will be lower Lipschitz for Lebesgue almost every $a \in S^{d-1}$, as proved in (Balan et al., 2022). In particular, it follows that $s_N$ is lower Lipschitz in expectation. As discussed in Subsection B.1, this implies that $s$ is lower Lipschitz in expectation as well. $\square$

### C.4 ANALYZING RELU

**Theorem 3.1.** *For $(n, d, p, \Omega, z)$ satisfying our standing assumptions, assume that $a \sim S^{d-1}$ and $b \sim [-B, B]$. Then $m_{ReLU}(\cdot; a, b)$ is uniformly Lipschitz. Moreover,*

1. *$m_{ReLU}(\cdot; a, b)$ is not $\alpha$ lower-Hölder in expectation for any $\alpha < \frac{p+1}{p}$.*

2. *If $\|x\| < B$ for all $x \in \Omega$, then $m_{ReLU}(\cdot; a, b)$ is $\frac{p+1}{p}$ lower-Hölder in expectation.*

*Proof.* We divide the proof into three parts, in accordance with the three parts of the theorem.

**Part 1: Uniform Lipschitz** We first prove a uniform Lipschitz bound for all balanced multisets $Y, Y'$ of cardinality $k \leq n$. Denote $X = \rho_{(z)}(Y), X' = \rho_{(z)}(Y')$. Then for every permutation $\tau \in S_n$,

$$|m_{ReLU}(Y;a,b) - m_{ReLU}(Y';a,b)| = |m_{ReLU}(X;a,b) - m_{ReLU}(X';a,b)|$$

$$= |\sum_{i=1}^{n} ReLU(ax_i - b) - ReLU(ax'_{\tau(i)} - b)|$$

$$\overset{(*)}{\leq} \sum_{i=1}^{n} |(ax_i - b) - (ax'_{\tau(i)} - b)|$$

$$= \sum_{i=1}^{n} |a \cdot (x_i - x'_{\tau(i)})|$$

$$\overset{(**)}{\leq} \sum_{i=1}^{n} \|a\|_2 \|x_i - x_{\tau(i)}\|_2$$

$$= \sum_{i=1}^{n} \|x_i - x_{\tau(i)}\|_2,$$

where (*) is because $ReLU$ is Lipschitz with constant 1, and (**) is from Cauchy-Schwartz.

Since the inequality we obtain holds for all permutations $\tau$, we can take the minimum over all permutations to obtain that

$$|m_{ReLU}(Y; a, b) - m_{ReLU}(Y'; a, b)| \leq W_1(X, X') = W_1^z(Y, Y')$$

To address multisets $Y, Y'$ of different sizes, we first note that since the elements of the multisets are in $\Omega$, and the parameters $a, b$ come from a compact set, there exists some constant $M > 0$ such that

$$|m_{ReLU}(Y; a, b) - m_{ReLU}(Y'; a, b)| \leq M.$$

On the other hand, for all $Y, Y'$ of different sizes, we will always have that

$$W_1^z(Y, Y') \geq dist(z, \Omega) > 0$$

therefore

$$|m_{ReLU}(Y; a, b) - m_{ReLU}(Y'; a, b)| \leq M \frac{W_1^z(Y, Y')}{W_1^z(Y, Y')} \leq \frac{M}{dist(z, \Omega)} W_1^z(Y, Y')$$

Combining this with our bound for multisets of equal cardinality, we see that $m_{ReLU}$ is uniformly Lipschitz with constant $\max\{1, \frac{M}{dist(z, \Omega)}\}$.

**Part 2: Lower bound on expected Hölder exponent**

We show that $m_{ReLU}$ is not $\beta$ lower-Hölder in expectation for all $\beta < (p+1)/p$.

Let $X$ be some matrix in $\Omega^n$ whose first two columns are the same $x_1 = x_2$. Let $q$ be a vector with unit norm. For every $\epsilon > 0$ define

$$X_\epsilon = [x_1 - \epsilon q, x_1 + \epsilon q, x_3, \ldots, x_n]$$

It is not difficult to see that for all small enough $\epsilon$ we have that $W_1(X_\epsilon, X) = 2\epsilon$. On the other hand, for every fixed $a \in S^{d-1}$ and $b \in [-B, B]$ we have that, denoting $y = a \cdot x_1$ and $\delta = |\epsilon a \cdot q|$, we have

$$\begin{aligned}
m_{ReLU}(X; a, b) - m_{ReLU}(X_\epsilon; a, b) &= 2\text{ReLU}(a \cdot x_1 - b) - \text{ReLU}(a \cdot (x_1 - \epsilon q) - b) \\
&\quad - \text{ReLU}(a(\cdot x_1 + \epsilon q) - b) \\
&= 2\text{ReLU}(y - b) - [\text{ReLU}(y + \delta - b) + \text{ReLU}(y - \delta - b)]
\end{aligned}$$

Note that if $b > y + \delta$ then the expression above will be zero because all arguments of the ReLUs will be negative, and if $b < y - \delta$ then the expression above will also be zero because the arguments of all ReLUs will be positive so that we obtain

$$2\text{ReLU}(y - b) - [\text{ReLU}(y + \delta - b) + \text{ReLU}(y - \delta - b)] = 2(y - b) - [y + \delta - b + y - \delta - b] = 0$$

Thus this expression will not vanish only if $b \in [y - \delta, y + \delta]$ which is an interval of diameter $2\delta \leq 2\epsilon$. For each $b$ in this interval we have, since ReLU is 1-Lipschitz, that

$$\begin{aligned}
|2\text{ReLU}(y - b) &- [\text{ReLU}(y + \delta - b) + \text{ReLU}(y - \delta - b)]| \leq \\
&|\text{ReLU}(y - b) - \text{ReLU}(y + \delta - b)| + |\text{ReLU}(y - b) - \text{ReLU}(y - \delta - b)| \leq 2\delta
\end{aligned}$$

So that in total the expectation for fixed $a$, over all $b$, is bounded by

$$\mathbb{E}_{b \sim [-B, B]} |m_{ReLU}(X; a, b) - m_{ReLU}(X_\epsilon; a, b)|^p \leq (2\epsilon)^p \mu[y - \delta, y + \delta] \leq \frac{1}{2B}(2\epsilon)^{p+1}$$

Which implies the same bound when taking the expectation over $a$ and $b$. Thus overall we obtain for all $\beta < (p-1)/p$ that

$$\mathbb{E}_{a,b} \left\{ \frac{|m_{ReLU}(X; a, b) - m_{ReLU}(X_\epsilon; a, b)|}{W_1(X, X_\epsilon)^\beta} \right\}^p \leq \frac{1}{2B} \frac{(2\epsilon)^{p+1}}{2^{\beta/p} \epsilon^{\beta \cdot p}} = \frac{2^{p+1}}{2B2^\beta} \epsilon^{p+1-\beta \cdot p} \overset{\epsilon \to 0}{\Rightarrow} 0$$

While if $m_{ReLU}$ were $\beta$ lower-Hölder in expectation this expression should have been uniformly bounded away from zero.

**Part 3: lower-Hölder in expectation** Next we show that $m_{ReLU}$ is $(p+1)/p$ lower-Hölder in expectation. We first consider the restriction of $m_{ReLU}$ to the subspace of $\mathcal{S}_{\leq n}(\Omega)$ which contains only multisets of cardinality exactly $n$. We denote this subspace by $\mathcal{S}_{=n}(\Omega)$. In this case, we realize

$m_{ReLU}$ as the composition of two functions: the functions $s(X; a) = sort(a^T X)$ from Lemma C.4, and the function $q(x; b) = \sum_i \text{ReLU}(x_i - b)$ Note that indeed $f(X; a, b) = (q \circ s)(a, b)$. Since we already know that $s$ is lower Lipschitz in expectation, it is sufficient to show that $q$ is $(p+1)/p$ lower-Hölder in expectation, due to the theorem on composition B.3.

Let us denote $\hat{B} = \max\{\|x\| \mid x \in \Omega\}$. By assumption $\hat{B} < B$.

Note that the domain of $q$ is contained in

$$\Omega_{[-B,B]} = \{x \in \mathbb{R}^n \mid -B \le x_1 \le x_2 \le \ldots \le x_n \le B\}.$$

Since the 2-norm and $\infty$-norm are equivalent on $\mathbb{R}^n$ C.1, to address the case of balanced multisets it is sufficient to prove

**Lemma C.5.** *Let $p > 0, B > 0$ and $n$ be a natural number. Let $q_b(x) = q(x; b)$ be the function described previously, defined on the domain*

$$\Omega_{[A,B]} = x \in \mathbb{R}^n \mid \quad A \le x_1 \le \ldots \le x_n \le B$$

*There is a constant*

$$C_{n,p} = \frac{1}{8n(B4 - A)^p}$$

*such that for all $x, y \in \Omega_{[A,B]}$,*

$$\mathbb{E}_{b \sim [A,B]} |q_b(x) - q_b(y)|^p \ge C_{n,p} \|x - y\|_\infty^{p+1}.$$

*Proof.* Let $x \ne y$ be a pair in $\Omega_{[A,B]}$. Let $s$ be an index for which $|y_s - x_s| = \|y - x\|_\infty$. without loss of generality assume that $y_s > x_s$. For every $b \in \mathbb{R}$ we denote $\Delta(b) = q_b(y) - q_b(x)$. Let $t$ be the smallest integer such that $x_t \ge y_s$. Note that $t > s$. We now have

$$\Delta(y_s) = \sum_{j=1}^n [\text{ReLU}(y_j - y_s) - \text{ReLU}(x_j - y_s)] = \sum_{j>s}(y_j - y_s) - \sum_{j \ge t}(x_j - y_s)$$

$$= \sum_{s<k<t}(y_k - y_s) + \sum_{j \ge t}(y_j - x_j)$$

Now

$$\Delta(x_s) = \sum_{i<s} \text{ReLU}(y_i - x_s) + (y_s - x_s) + \sum_{j>s}[(y_j - x_s) - (x_j - x_s)]$$

$$\ge (y_s - x_s) + \sum_{j>s}(y_j - x_j)$$

$$= (y_s - x_s) + \sum_{s<k<t}(y_k - y_s) + \sum_{s<k<t}(y_s - x_k) + \sum_{j \ge t}(y_j - x_j)$$

$$\ge (y_s - x_s) + \sum_{s<k<t}(y_k - y_s) + \sum_{j \ge t}(y_j - x_j)$$

$$= (y_s - x_s) + \Delta(y_s)$$

We deduce that $\Delta(x_s) - \Delta(y_s) \ge y_s - x_s$, and therefore at least one of $|\Delta(x_s)|$ and $|\Delta(y_s)|$ is larger than $\frac{y_s - x_s}{2}$, so we found some $b_0$ for which $|\Delta(b_0)| \ge \frac{y_s - x_s}{2}$. Next, we note that since $\Delta$ is a sum of $2n$ ReLU functions which are all 1-Lipschitz, $\Delta$ is $(2n)$ Lipschitz. Therefore if $b$ is such that $|b - b_0| \le \delta := \frac{1}{8n}(y_s - x_s)$ then

$$||\Delta(b)| - |\Delta(b_0)|| \le |\Delta(b) - \Delta(b_0)| \le 2n\delta = \frac{y_s - x_s}{4}$$

implying that $|\Delta(b)| \ge \frac{y_s - x_s}{4}$. Thus

$$\mathbb{E}_b |q_b(y) - q_b(x)|^p = \mathbb{E}_b |\Delta(b)|^p \ge \mu\{b \mid |b - b_0| < \delta\} \cdot \left[\frac{y_s - x_s}{4}\right]^p \ge \frac{\delta}{(B - A)4^p} |y_s - x_s|^p$$

$$= \frac{1}{8n(B - A)4^p} |y_s - x_s|^{p+1} = \frac{1}{8n(B - A)4^p} \|y - x\|_\infty^{p+1}$$

$\square$

Now let us consider the case of unbalanced multisets $Y, Y'$ in $\mathcal{S}_{\leq n}(\Omega)$. Our goal will be to show that the distance between all unbalanced multisets is uniformly bounded from below away from zero.

For fixed $a \in S^{d-1}$ denote $y = a \cdot Y$ and $y' = a \cdot Y'$. Denote $x = \rho_{(-B)}(y)$, that is, $x$ is the multiset obtained from adding elements with value $-B$ to $y$ until it has $Y$ elements. Similarly, denote $x' = \rho_{(-B)}(y')$. Note that since $y$ and $y'$ don't have the same number of elements, and all entries of $y$ are in $[-B', B']$, we have that

$$\|x - x'\|_\infty \geq B - B'.$$

For all $b \in [-B, B]$ we have that $\mathrm{ReLU}(-B - b) = 0$, and therefore, according to Lemma C.5, we have

$$\mathbb{E}_b |m_{ReLU}(Y; a, b) - m_{ReLU}(Y'; a, b)|^p = \mathbb{E}_b |q_b(y) - q_b(y')|^p$$
$$= \mathbb{E}_b |q_b(x) - q_b(x')|^p \geq C_{n,p} \|x - x'\|_\infty^{p+1} \geq C_{n,p}(B - B')^{p+1}$$

We deduce that

$$\mathbb{E}_{a,b} |m_{ReLU}(Y; a, b) - m_{ReLU}(Y'; a, b)|^p \geq C_{n,p}^p (B - B')^p$$
$$= C_{n,p}^p (B - B')^p \left( \frac{W_1^z(Y, Y')}{W_1^z(Y, Y')} \right)^{p+1} \geq C \left( W_1^z(Y, Y') \right)^{p+1}$$

For an appropriate constant $C$, where we use the fact that $W_1^z$ is bounded from above. We have obtained a $(p + 1)/p$ lower Hölder bound for both balanced and unbalanced multisets, and so we are done. $\square$

**Adaptive ReLU**

**Theorem 3.2.** *For $(n, d, p, \Omega, z)$ satisfying our standing assumptions, assume that $a \sim S^{d-1}$ and $t \sim [0, 1]$. Then the function $m_{\mathrm{ReLU}}^{\mathrm{adapt}} : \mathcal{S}_{\leq n}(\Omega) \times S^{d-1} \times [0, 1] \to \mathbb{R}^4$ is uniformly Lipschitz and $\frac{p+1}{p}$ lower-Hölder in expectation. Moreover, when $n \geq 4$, the function $m_{\mathrm{ReLU}}^{\mathrm{adapt}}$ is not $\alpha$ lower-Hölder in expectation for any $\alpha < \frac{p+1}{p}$.*

*Proof.* To prove this theorem, we first recall the definition of $m_{\mathrm{ReLU}}^{\mathrm{adapt}}$

$$m = \min\{a \cdot x_1, \ldots, a \cdot x_r\}, \quad M = \max\{a \cdot x_1, \ldots, a \cdot x_r\}, \quad b = (1 - t)m + tM$$

$$m_{\mathrm{ReLU}}^{\mathrm{adapt}}(X; a, t) = [r, m, M, \frac{1}{r} \sum_{i=1}^r \mathrm{ReLU}(a \cdot x_i - b)]$$

To begin with, we note that the case of unbalanced multisets is easy to deal with. There exists constants $0 < c = 1 < C$ such that, for every pair of unbalanced multisets $Y, Y'$, and every choice of parameters $a, t$,

$$1 \leq \|m_{\mathrm{ReLU}}^{\mathrm{adapt}}(X; a, t) - m_{\mathrm{ReLU}}^{\mathrm{adapt}}(X; a, t)\|_p \leq C$$

The lower bound follows from the fact that the first coordinate of $m_{\mathrm{ReLU}}^{\mathrm{adapt}}$ is the cardinality of the sets. The upper bound follows from compactness. Similarly, the augmented Wasserstein distance between all unbalanced multisets in $\mathcal{S}_{\leq n}(\Omega)$ is uniformly bounded from above and below. This can be used to obtain both uniform upper Lispchitz bounds, and lower Hölder bounds, as discussed in previous proofs.

Thus, it is sufficient to prove uniform upper Lispchitz bounds, and lower Hölder bounds in expectation, for balanced multisets. Without loss of generality we can assume the balanced multisets both have maximal cardinality $n$. So we need to prove the claim on the space $\mathcal{S}_{=n}(\Omega)$.

We can write $m_{\mathrm{ReLU}}^{\mathrm{adapt}}$, restricted to $\mathcal{S}_{=n}(\Omega)$, as a composition

$$m_{\mathrm{ReLU}}^{\mathrm{adapt}}(X; a, t) = Q \circ s(X; a.t)$$

where $s(X; a) = \mathrm{sort}(a^T X)$, which is the uniformly upper Lipschitz, and lower Lipschitz in expectation function defined in Lemma C.4, and $Q(x; t)$ is defined via

$$m = \min\{x_1, \ldots, x_n\}, \quad M = \max\{x_1, \ldots, x_n\}, \quad b = (1 - t)m + tM$$

and

$$Q(x;t) = [n, m, M, \frac{1}{n}\sum_{i=1}^{n}\text{ReLU}(x_i - b)]$$

As $s$ is is uniformly upper Lipschitz, and lower Lipschitz in expectation, it is sufficient to show that $Q$ is upper Lipchitz, and lower Hölder in expectation.

We note that since $\Omega$ is bounded, there exists some $B > 0$ such that $\|x\| \leq B, \forall x \in \Omega$, and for this $B$ we have that the image of $s$ is contained in

$$\Omega_{[-B,B]} = \{x \in \mathbb{R}^n | \quad -B \leq x_1 \leq \ldots \leq x_n \leq B\}$$

We can therefore think of $\Omega_{[-B,B]}$ as the domain of $Q$.

We begin with the upper Lipschitz bound. Let $x, y$ be vectors in $\Omega_{[-B,B]}$, which we identify with multisets with $n$ elements in $\mathbb{R}$. Denote the maximum and minimum of $x$ by $M_x$ and $m_x$. Define the maximum and minimum of $y$ by $M_y$ and $m_y$. Denote

$$\phi(s, t, m, M) = \text{ReLU}(s - [(1-t)m + tM])$$

Then for all $t \in [0, 1]$.

$$\begin{aligned}
\|Q(X;t) - Q(Y;t)\|_p &\leq C \cdot \|Q(X;t) - Q(Y;t)\|_1 \\
&= C \cdot (|m_x - m_y| + |M_x - M_y| + \frac{1}{n}|\sum_{i=1}^{n}\phi(x_i, t, m_x, M_x) - \sum_{i=1}^{n}\phi(y_i, t, m_y, M_y)|) \\
&\leq C \cdot (|m_x - m_y| + |M_x - M_y| + \frac{1}{n}\sum_{i=1}^{n}|x_i - y_i + t(M_x - M_x) + (1-t)(m_x - m_y)|) \\
&\leq C \cdot (|m_x - m_y| + |M_x - M_y| + \frac{1}{n}\sum_{i=1}^{n}(|x_i - y_i| + |(M_x - M_y)| + |(m_x - m_y))|) \\
&\leq C \cdot (\|x - y\|_\infty + \|x - y\|_\infty + \frac{1}{n}\sum_{i=1}^{n}3 \cdot \|x - y\|_\infty) \\
&= 5C \cdot \|x - y\|_\infty \leq 5CC' \cdot \|x - y\|_p
\end{aligned}$$

Where $C, C'$ are the constants obtained from norm equivalence over $\mathbb{R}^n$.

To obtain a Hölder lower bound, the idea of the proof is that for given balanced multisets $x, y \in \mathbb{R}^n$, we know from the analysis of $m_{\text{ReLU}}$ that we can get a lower-Hölder bound when considering biases going between

$$m_{x,y} = \min\{m_x, m_y\} \text{ and } M_{x,y} = \max\{M_x, M_y\}$$

and then showing that the difference between this case and the function $Q$ where the bias range depends on the maximum and minimum of the individual multisets $x, y$, is proportional to the different between the minimum and maximum of $x$ and $y$, which also appear in $Q$. Indeed, since ReLU is Lipschitz we have for every $s, m, M, \hat{m}, \hat{M}$ in $\mathbb{R}$ and $t \in [0, 1]$ that

$$\begin{aligned}
|\phi(s, t, m, M) - \phi(s, t, \hat{m}, \hat{M})| &\leq |-t(M - \hat{M}) - (1-t)(m - \hat{m})| \quad\quad (6) \\
&\leq t|M - \hat{M}| + (1-t)|m - \hat{m}| \leq |M - \hat{M}| + |m - \hat{m}|
\end{aligned}$$

Next, to bound $\|Q(x;t) - Q(y;t)\|_p$, due to equivalence of norms, it is sufficient to bound $\|Q(x;t) - Q(y;t)\|_1$. We obtain

$$\|Q(x;t) - Q(y;t)\|_1 = |m_x - m_y| + |M_x - M_y|$$

$$+ \frac{1}{n}|\sum_{i=1}^{n} \phi(x_i, t, m_x, M_x) - \sum_{i=1}^{n} \phi(y_i, t, m_y, M_y)|$$

$$= |m_x - m_{xy}| + |M_x - M_{xy}| + |m_{xy} - m_y| + |M_{xy} - M_y|$$

$$+ \frac{1}{n}|\sum_{i=1}^{n} \phi(x_i, t, m_x, M_x) - \sum_{i=1}^{n} \phi(y_i, t, m_y, M_y)|$$

$$\overset{(6)}{\geq} \frac{1}{n}|\sum_{i=1}^{n} \phi(x_i, t, m_{xy}, M_{xy}) - \sum_{i=1}^{n} \phi(x_i, t, m_x, M_x)|$$

$$+ \frac{1}{n}|\sum_{i=1}^{n} \phi(y_i, t, m_y, M_y) - \sum_{i=1}^{n} \phi(y_i, t, m_{xy}, M_{xy})|$$

$$+ \frac{1}{n}|\sum_{i=1}^{n} \phi(x_i, t, m_x, M_x) - \sum_{i=1}^{n} \phi(y_i, t, m_y, M_y)|$$

$$\overset{\text{triangle ineq.}}{\geq} \frac{1}{n}|\sum_{i=1}^{n} \phi(x_i, t, m_{xy}, M_{xy}) - \sum_{i=1}^{n} \phi(x_i, t, m_x, M_x)$$

$$+ \sum_{i=1}^{n} \phi(y_i, t, m_y, M_y) - \sum_{i=1}^{n} \phi(y_i, t, m_{xy}, M_{xy})$$

$$+ \sum_{i=1}^{n} \phi(x_i, t, m_x, M_x) - \sum_{i=1}^{n} \phi(y_i, t, m_y, M_y)|$$

$$= \frac{1}{n}|\sum_{i=1}^{n} \phi(x_i, t, m_{xy}, M_{xy}) - \sum_{i=1}^{n} \phi(y_i, t, m_{xy}, M_{xy})|$$

$$= |q(x; (1-t)m_{xy} + tM_{xy}) - q(y; (1-t)m_{xy} + tM_{xy})|$$

By Lemma C.5, we deduce that

$$\mathbb{E}_{t\sim[0,1]}\|Q(x;t) - Q(y;t)\|_1^p \geq \mathbb{E}_{b\sim[m_{xy}, M_{xy}]}|q(x;b) - q(y;b)|^p$$

$$\geq \frac{1}{8n4^p(M_{xy} - m_{xy})}\|y - x\|_\infty^{p+1} \geq \frac{1}{16nB4^p}\|y - x\|_\infty^{p+1}$$

by invoking the equivalence of the infinity norm and $p$ norm we are done.

Finally, we will show that, when $n \geq 4$, adaptive ReLU cannot be $\alpha$-Hölder for any $\alpha > (p+1)/p$. This argument is essentially a reduction to our argument in the standard ReLU summation case. For simplicity of notation we prove this for the case where $d = 1$. Extending the argument to the $d \geq 1$ case is straightforward.

For any $\epsilon > 0$, consider the sets

$$X = \{\!\{0, 0, 1, -1\}\!\}, \quad X_\epsilon = \{\!\{\epsilon, -\epsilon, 1, -1\}\!\}.$$

and note that

$$W_1(X, X_\epsilon) = 2\epsilon.$$

Next, we note that for all $a$ in the zero dimensional unit circle $\{-1, 1\}$ we have that $a \cdot X = X, a \cdot X_\epsilon = X_\epsilon$, and moreover, that $X$ and $X_\epsilon$ have the same number of elements. Therefore

$$\mathbb{E}_{a,t}\|m_{\text{ReLU}}^{\text{adapt}}(X_\epsilon; a, t) - m_{\text{ReLU}}^{\text{adapt}}(X; a, t)\|_p = \frac{1}{4}\mathbb{E}_{b\sim[-1,1]}|\text{ReLU}(-\epsilon-b) + \text{ReLU}(\epsilon-b) - 2\text{ReLU}(-b)|.$$

Next, we note, as in previous arguments, that

$$|\text{ReLU}(-\epsilon - b) + \text{ReLU}(\epsilon - b) - 2\text{ReLU}(-b)|^p = 0, \quad \forall b \notin [-\epsilon, \epsilon]$$

$$|\text{ReLU}(-\epsilon - b) + \text{ReLU}(\epsilon - b) - 2\text{ReLU}(-b)|^p < (2\epsilon)^p, \quad \forall b \in \mathbb{R}$$

Thus, if $b$ is in $[-\epsilon, \epsilon]$, which occurs with probability $\epsilon$, the expression above will be at most $(2\epsilon)^p$, and otherwise it will be zero. The expectation of this expression over $b$ is thus bounded by $2^p \epsilon^{p+1}$. Piecing all this together, we obtain for all $\beta < (p+1)/p$

$$\mathbb{E}_{a,t}\left[\frac{\|m_{\mathrm{ReLU}}^{\mathrm{adapt}}(X_\epsilon; a, t) - m_{\mathrm{ReLU}}^{\mathrm{adapt}}(X; a, t)\|_p}{W_1^\beta(X, X_\epsilon)}\right]^p \leq \frac{1}{4^p}\frac{2^p \epsilon^{p+1}}{2^{p\cdot\beta}\epsilon^{p\cdot\beta}} = 2^{p(1-2-\beta)}\epsilon^{p(\frac{p+1}{p}-\beta)} \overset{\epsilon \to 0}{\to} 0.$$

This shows that $m_{\mathrm{ReLU}}^{\mathrm{adapt}}$ is not $\beta$ lower-Hölder in expectation.

$\square$

**Remark C.6.** The last part of the proof above can be used to derive adversarial examples on which adaptive-ReLU will have bad distortion: namely,

$$X = \{\!\!\{0, 0, 1, -1\}\!\!\}, \quad X_\epsilon = \{\!\!\{\epsilon, -\epsilon, 1, -1\}\!\!\}.$$

On these examples, adaptive-ReLU and ReLU summation will both encounter high distortion. In contrast, the examples discussed in the main text, like

$$Y = \{\!\!\{0, 0\}\!\!\}, Y_\epsilon = \{\!\!\{\epsilon, -\epsilon\}\!\!\}$$

one can verify directly that the expectation of $|m_{\mathrm{ReLU}}^{\mathrm{adapt}}(Y', a, t) - m_{\mathrm{ReLU}}^{\mathrm{adapt}}(Y, a, t)|^p$ scales linearly in $\epsilon^p$. This is because the bias for these examples naturally is in the range $[-\epsilon, \epsilon]$ where 'it can make a difference', while in the $X, X_\epsilon$ example, the bias is chosen from all of $[-1, 1]$ and its probability to land in the domain $[-\epsilon, \epsilon]$ where 'it can make a difference' scales like $\epsilon$.

We believe this gives a good intuition for the reason why adaptive-ReLU is successful on many of the adversarial examples illustrated in the text, and also suggests how they can be changed so that adaptive ReLU will fail: we simply need to take these examples and add to all multisets considered a large positive and a large negative element. An example of this idea is shown in Figure 7, where adaptive-ReLU is initially successful in a classification task based on adversarial examples (subplot (a)), but fails completely once a large positive and negative element are added (subplot (b)).

# D    LIPSCHITZ COMBINE OPERATIONS

In this section we describe how to construct COMBINE functions which are both uniformly Lipschitz, and lower-Lipschitz in expectation. The input to these functions are a pair of vectors $x_1, y_1$ with the metric $\|x_1\|_p + \|y_1\|_p$. The output will be a vector (or scalar), and the metric on the output space will again be the $p$ norm.

Our analysis covers four 2-tuple embeddings, all of which are uniformly upper Lipschitz, and lower Lipschitz in expectation.

**Theorem D.1.** *Let $p > 0$, the following 2-tuple embeddings are all bi-Lipschitz in expectation with respect to $l_p$ and the 2-tuple metric previously defined.*

1. *Linear combination: Given a 2-tuple $(x, y) \in \mathbb{R}^d \times \mathbb{R}^d$ and $\alpha \sim U[-D, D]$ we define*

$$f(x, y; \alpha) = \alpha \cdot x + y$$

2. *Linear transform and sum: Given a 2-tuple $(x, y) \in \mathbb{R}^k \times \mathbb{R}^l$, and $A \in \mathbb{R}^{l \times k}$, whose rows are independently and uniformly sampled from $S^{k-1}$, we define*

$$f(x, y; A) = Ax + y$$

3. *Concatenation: This embedding concatenates the tuple entries, and has no parameters*

$$f(x, y) = \begin{bmatrix} x \\ y \end{bmatrix}$$

4. *Concat and project: Given $(x, y) \in \mathbb{R}^k \times \mathbb{R}^l$ and $\theta \sim S^{k+l-1}$*

$$f(x, y; \theta) = \theta \cdot \begin{bmatrix} x \\ y \end{bmatrix}$$

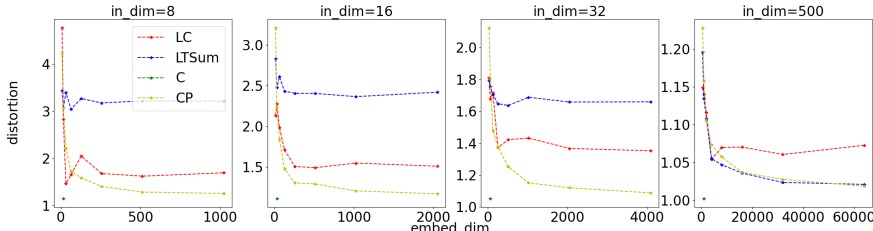

Figure 4: Distortion of the 2-tuple embeddings as a function of input and embedding dimension. LC=linear combination, LTSum=linear transform and sum, CP=concat project, C=concatenation.

We note that the last method, concat and project, is the method used in the definition of SortMPNN in the main text. We also note that the first method, linear combination, is the method used in (Xu et al., 2019). Unlike the other methods, it requires the vectors $x, y$ to be of the same dimension.

In order to compare the proposed COMBINE functions which all share the bi-Lipschitz in expectation property, one could further explore separation quality by comparing the distortion defined by $\frac{M}{m}$ where $M$ ($m$) is the upper (lower) Lipschitz in expectation bound. A lower distortion would indicate that the function doesn't change the input metric as much (possibly up to multiplication by some constant), resulting in high quality separation.

In figure 4, we plot the empirical distortion of the different COMBINE functions with varying input and embedding dimensions. The embedding dimension is controlled by stacking multiple instances of the functions with independent parameters. Note that *concatenation* isn't parametric and therefore we only have a single output dimension for it.

The experiment is run on $1,000$ different random tuple pairs $\{(x, y), (z, w)\}$ for each of the four input dimensions experimented with, where $x, y, z, w \in \mathbb{R}^{\text{in\_dim}}$. All random data vectors are sampled with entries drawn from the normal distribution. The empirical distortion is then computed by taking the largest empirical ratio $M$ between the input and output distances and dividing it by the smallest ratio $m$ between the input and output distances.

As we see in the figure, all proposed functions stabilize around a constant value as the embedding dimension increases. This value is the expected distortion on the experiment data. We can see that *concat and project* maintains low expected distortion across all settings, while *linear combination* has a higher expected distortion but not by much. *Linear transform and sum* seems to have higher expected distortion for lower input dimensions, but it improves with large input dimensions, matching *concat and project* distortion.

Another interesting result is that we can see that *concat and project* seems to have the highest variance of the three for most input dimensions, since the empirical distortion for low embedding dimensions is higher than other functions in most cases.

*Proof of Theorem D.1.*

**Linear combination**: Let $x, z \in \mathbb{R}^k, y, w \in \mathbb{R}^l$ We begin by proving that $f$ is lower Lipschitz in expectation.

$$\mathbb{E}_{\alpha \sim U[-D,D]}[||f(x, y; \alpha) - f(z, w; \alpha)||_p^p] = \mathbb{E}_{\alpha \sim U[-D,D]}[||(\alpha \cdot x + y) - (\alpha \cdot w + z)||_p^p]$$

$$= \mathbb{E}_{\alpha \sim U[-D,D]}[||\alpha \cdot (x - z) + (y - w)||_p^p]$$

From norm equivalence C.1, $\exists c_1$ s.t.

$$\geq c_1 \cdot \mathbb{E}_{\alpha \sim U[-D,D]}[||\alpha \cdot (x - z) + (y - w)||_1^p]$$

$$\overset{\text{Jensen's ineq.}}{\geq} c_1 \cdot \mathbb{E}_{\alpha \sim U[-D,D]}[||\alpha \cdot (x - z) + (y - w)||_1]^p$$

$$= c_1 \cdot E_{\alpha \sim U[-D,D]}[||\alpha \cdot (x - z)||_1 + ||(y - w)||_1]^p$$

From norm equivalence C.1, $\exists c_2$ s.t.

$$\geq c_1 \cdot c_2 \cdot (E_{\alpha \sim U[-D,D]}[|\alpha|] \cdot ||(x - z)||_p + ||(y - w)||_p)^p$$

$$= c_1 \cdot c_2 \cdot (\frac{D}{2} \cdot ||(x-z)||_p + ||(y-w)||_p)^p$$

$$\geq c_1 \cdot c_2 \cdot min(1, \frac{D}{2}^p) \cdot (||(x-z)||_p + ||(y-w)||_p)^p$$

For uniform upper Lipschitz bounds we have

$$||f(x,y;\alpha) - f(z,w;\alpha)||_p = ||(\alpha \cdot x + y) - (\alpha \cdot w + z)||_p$$
$$\leq ||\alpha \cdot x - \alpha \cdot w||_p + ||y - z||_p$$
$$\leq \max(1, D)^p \cdot (||(x-z)||_p + ||(y-w)||_p)$$

**Linear transform and sum**: We prove lower-Lipschitzness in expectation. Let $a_i$ denote the i'th row of $A$. Recall that for any $1 \leq i \leq l$, $a_i$ is drawn uniformly from $S^{k-1}$. Let $x, z \in \mathbb{R}^k, y, w \in \mathbb{R}^l$

$$\mathbb{E}_A[||f(x,y;A) - f(z,w;A)||_p^p] = \mathbb{E}_A[||A(x-z) + (y-w)||_p^p]$$

$$= \mathbb{E}_A[\sum_{i=1}^{l} |a_i \cdot (x-z)|^p + \sum_{j=1}^{l} |y_j - w_j|^p] = \sum_{i=1}^{l} \mathbb{E}_{a_i \sim S^{k-1}}[|a_i \cdot (x-z)|^p] + \sum_{j=1}^{l} |y_j - w_j|^p$$

From lemma C.3 $\exists b > 0$ s.t.

$$\mathbb{E}_A[||f(x,y;A) - f(z,w;A)||_p^p] \geq n \cdot b \cdot ||x-z||_p^p + ||y-w||_p^p$$

Let us denote $t = \begin{bmatrix} ||x-z||_p \\ ||y-w||_p \end{bmatrix} \in \mathbb{R}^2$. Then

$$n \cdot b \cdot ||x-z||_p^p + ||y-w||_p^p \geq min(1, n \cdot b) \cdot (||x-z||_p^p + ||y-w||_p^p) = min(1, n \cdot b) \cdot ||t||_p^p$$

From norm equivalence C.1, $\exists c > 0$

$$\geq min(1, n \cdot b) \cdot c \cdot ||t||_1^p = min(1, n \cdot b) \cdot c \cdot (||x-z||_p + ||y-w||_p)^p$$

Uniform upper Lipschitzness is straightforward to prove.

**Concatenation**: We show that this non-parametric function is bi-Lipschitz. Let $x, z \in \mathbb{R}^k, y, w \in \mathbb{R}^l$. Denote as before $t = \begin{bmatrix} ||x-z||_p \\ ||y-w||_p \end{bmatrix} \in \mathbb{R}^2$. Then the distance difference in the output space is given by $||f(x,y) - f(z,w)||_p^p = ||t||_p$, while the difference in the input space is given by

$$||x-z||_p + ||y-w||_p = ||t||_1.$$

The claim follows from equivalence of $p$ norm and $1$ norm on $\mathbb{R}^2$ C.1.

**Concat and project**: The claim follows from the fact that the concatenation operation is bi-Lipschitz, and Lemma C.3. □

# E   TREE MOVER'S DISTANCE

In this appendix section we review the definition of the Tree Mover's Distance (TMD) from (Chuang & Jegelka, 2022), which is the way we measures distances between graphs in this paper.

We first review Wasserstein distances. Recall that if $(X, D)$ is a metric space, $\Omega \subseteq X$ is a subset, and $z$ is some point in $X$ with $dist(z, \Omega) > 0$, then we can define the Wasserstein distance on the space of multisets consisting of $n$ elements in $\Omega$ via

$$W_1(\{\!\{x_1, \ldots, x_n\}\!\}, \{\!\{y_1, \ldots, y_n\}\!\}) = \min_{\tau \in S_n} \sum_{j=1}^{n} D(x_j, y_{\tau(j)})$$

The augmentation map on multisets of size $r \leq n$ is defined as

$$\rho_{(z)}(\{\!\{x_1, \ldots, x_r\}\!\}) = \{\!\{x_1, \ldots, x_r, x_{r+1} = z, \ldots, x_n = z\}\!\}$$

and the augmented distance on multisets of size up to $n$ is defined via

$$W_{D,1}^{(z)}(X, \hat{X}) = W_{D,1}(\rho_{(z)}X, \rho_{(z)}\hat{X})$$

We now return to define the TMD. We consider the space of graphs $G_{\leq n}(\Omega)$, consisting of graphs with $\leq n$ nodes, with node features coming from a compact domain $\Omega \subseteq \mathbb{R}^d$. We also fix some $z \in \mathbb{R}^d \setminus \Omega$. The TMD is defined using the notion of computation trees:

**Definition E.1.** (Computation Trees). Given a graph $G = (V, E)$ with node features $\{\!\{x_v\}\!\}_{v \in V}$, let $T_v^{(0)}$ be the rooted tree with a single node $v$, which is also the root of the tree, and node features $x_v$. For $K \in \mathbb{N}^+$ let $T_v^{(K)}$ be the depth-$K$ computation tree of node $v$ constructed by connecting the neighbors of the leaf nodes of $T_v^{(K-1)}$ to the tree. Each node is assigned the same node feature it had in the original graph $G$. The multiset of depth-$K$ computation trees defined by $G$ is denoted by $\mathcal{T}_G^{(K)} := \{\!\{T_v^{(K)}\}\!\}_{v \in V}$. Additionally, for a tree $T_r$ with root $r$, we denote by $\mathcal{T}_r$ the multiset of subtrees that root at the descendants of $r$.

**Definition E.2.** (Blank Tree). A blank tree $\bar{T}_z$ is a tree (graph) that contains a single node and no edge, where the node feature is the blank vector $z$.

Recall that by assumption, all node features will come from the compact set $\Omega$, and $z \notin \Omega$.

We can now define the tree distance:

**Definition E.3.** (Tree Distance).[4] The distance between two trees $T_a, T_b$ with features from $\Omega$ and $z \notin \Omega$, is defined recursively as

$$TD(T_a, T_b) := \begin{cases} \|x_a - x_b\|_p + W_{TD,1}^{(\bar{T}_z)}(\mathcal{T}_a, \mathcal{T}_b) & \text{if } K > 0 \\ \|x_a - x_b\|_p & \text{otherwise} \end{cases}$$

where $K$ denotes the maximal depth of the trees $T_a$ and $T_b$.

**Definition E.4.** (Tree Mover's Distance). Given two graphs, $G, H$ and $w, K \geq 0$, the tree mover's distance is defined as

$$TMD^{(K)}(G, H) = W_{TD,1}^{(\bar{T}_z)}(\mathcal{T}_G^{(K)}, \mathcal{T}_H^{(K)})$$

where $\mathcal{T}_G^{(K)}$ and $\mathcal{T}_H^{(K)}$ denote the multiset of all depth $K$ computational trees arising from the graphs $G$ and $H$, respectively. We refer the reader to (Chuang & Jegelka, 2022) where they prove this is a pseudo-metric that fails to distinguish only graphs which cannot be separated by $K$ iterations of the WL test.

## F  MPNN HÖLDER PROOFS

In the main text we informally stated the following theorem

**Theorem 4.1.** *(Uniformly Lipschitz MPNN embeddings, informal version) Let $f : G_{\leq n}(\Omega) \to \mathbb{R}^m$ be an MPNN with $K$ layers. If the functions used for the aggregation $\phi^{(k)}$, combine $\psi^{(k)}$, and readout $\eta$ are all uniformly upper Lipschitz, then $f$ is uniformly upper Lipschitz with respect to $TMD^{(K)}$. In particular, ReluMPNN, SmoothMPNN, AdaptMPNN and SortMPNN are all uniformly upper Lipschitz.*

Before stating the formal theorem and proof, we will present an important lemma alongside some assumptions and notation that will be used throughout this section.

**Lemma F.1.** *Let $\Omega \subseteq \mathbb{R}^d$ be a compact set and let $f : G_{\leq n}(\Omega) \to \mathbb{R}^m$ be a depth $K$ MPNN such that all the aggregation $\phi^{(k)}$, combine $\psi^{(k)}$ and readout $\eta$ are continuous. Let $z_k \notin \Omega_k$, where $\Omega_k$ is the set of all possible feature vectors after running $k \leq K$ message passing layers. Then, for any $G \in G_{\leq n}(\Omega), v \in V_G$ and for any $p \geq 1$, there exist $C \geq c > 0$ such that*

$$C \cdot TD(T_v^{(k)}, \bar{T}_z) \geq ||z_k - x_v^{(k)}||_p \geq c \cdot TD(T_v^{(k)}, \bar{T}_z)$$

---

[4]Note the difference from the original definition in (Chuang & Jegelka, 2022) is due to our choice to set the depth weight to 1 and using the 1-Wasserstein which is equivalent to optimal transport

*Proof.* We begin by showing that $\Omega_k$ is compact.

Since the number of nodes in graphs from $G_{\leq n}(\Omega)$ is bounded, the set of all possible graph topologies, i.e. pairs $(V, E)$ with at most $n$ nodes, is finite. Let $(V, E)$ be such a fixed topology. For every $k$, the function $f_{V,E}^{(k)}((x_v)_{v \in \Omega}, \theta)$ mapping features $\Omega$ and parameters $\theta$ to new parameter $x_v^{(k)}$ is continuous since the aggregation and combine functions are continuous. Accordingly the image $f(k)_{V,E}(\Omega)$ is compact, and therefore

$$\Omega_k = \bigcup_{(V,E),|V| \leq n} f_{V,E}^{(k)}(\Omega)$$

is compact as well.

Therefore:

$$a := \inf_{x \in \Omega_k} ||x - z_k||_p > 0$$

$$d := \sup_{x \in \Omega_k} ||x - z_k||_p < \infty$$

Additionally, since $\Omega$ is compact and $z \notin \Omega$, we can show recursively, starting from $k = 1$, that for all $k \in \{0 \ldots, K\}$ and all trees $T_v^{(k)} \in \mathcal{T}_{G_{\leq n}(\Omega)}^{(k)}$,

$$b := \inf_{T_v^{(k)} \in \mathcal{T}_{G_{\leq n}(\Omega)}^{(k)}} TD(\bar{T}_z, T_v^{(k)}) > 0$$

$$e := \sup_{T_v^{(k)} \in \mathcal{T}_{G_{\leq n}(\Omega)}^{(k)}} TD(\bar{T}_z, T_v^{(k)}) < \infty$$

Where $\mathcal{T}_{G_{\leq n}(\Omega)}^{(k)}$ denotes the set of all possible height-$k$ computation trees from $G_{\leq n}(\Omega)$. Therefore, $\forall G \in G_{\leq n}(\Omega), v \in V_G$:

$$\frac{d}{b} \cdot TD(\bar{T}_z, T_v^k) \geq d \cdot \frac{b}{b} = d \geq ||z_k - x_v^{(k)}||_p \geq a = a \cdot \frac{e}{e} \geq \frac{a}{e} \cdot TD(\bar{T}_z, T_v^k)$$

To conclude we define $C := \frac{d}{b}, c := \frac{a}{e}$.

$\square$

**Assumptions and notations** As stated in the main text, we consider graphs with up to $n$ nodes. We will also make the disjointness assumption: We assume as previously that the initial features of the graphs all reside in a compact set $\Omega = \Omega_0 \subseteq \mathbb{R}^d$. We denote by $\Omega_k$ the space of all possible features which can be obtained by the MPNN at question after $k$ iteration, with any choice of parameters. This set is also compact (see proof of F.1). The disjointness assumptions is that the 'augmentation vector' $z_k$ is not in $\Omega_k$, for all $k = 0, 1, \ldots, K$.

We define for $k = 0, \ldots, K - 1$

$$x_{v,s}^{(k)} = \begin{cases} x_s^{(k)} & \text{if } (v,s) \in E \\ z_k & \text{if } otherwise \end{cases}, \text{ and } T_{v,s}^{(k)} = \begin{cases} T_s^{(k)} & \text{if } (v,s) \in E \\ \bar{T}_z & \text{if } otherwise \end{cases}$$

For $K$ we denote

$$x_s^{(K)} = \begin{cases} x_s^{(K)} & \text{if } s \in V \\ z_K & \text{if } otherwise \end{cases}, \text{ and } T_s^{(K)} = \begin{cases} T_s^{(K)} & \text{if } s \in V \\ \bar{T}_z & \text{if } otherwise \end{cases}$$

We will also denote

$$x_{v,\bullet}^{(k)} = [x_{v,1}^{(k)}, \ldots, x_{v,n}^{(k)}] \text{ and } x_{\bullet}^{(K)} = [x_1^{(K)}, \ldots, x_n^{(K)}]$$

We can now state the full formal statement of the theorem:

**Theorem F.2.** *(Uniformly Lipschitz MPNN embeddings, full version) Let $p \geq 1$ and $K \in \mathbb{N}$, Let $f : G_{\leq n}(\Omega) \to \mathbb{R}^m$ be a continuous MPNN with $K$ message passing layers. Under the above assumptions F and given the following holds for all $1 \leq k \leq K$:*

1. *The aggregation, $\phi^{(k)}$, is uniformly upper Lipschitz w.r.t. the augmented Wasserstein distance $W_1^{(z_k)}$ on $\mathcal{S}_{\leq n}(\Omega_k)$.*

2. *The combine function $\psi^{(k)}$, is uniformly upper Lipschitz.*

3. *the readout function $\eta$, is uniformly upper Lipschitz w.r.t. the augmented Wasserstein distance $W_1^{z^{(K+1)}}$ on $\mathcal{S}_{\leq n}(\Omega_{K+1})$..*

*Then, there exist constants $C_1, C_2 > 0$ such that*

1. *The **node embeddings** after $k$ message passing layers are uniformly upper Lipschitz w.r.t. $TD$ on their depth $k$ computation trees*

$$\|x_v^{(k)} - x_u^{(k)}\|_p^p \leq C_1 \cdot TD^p(T_v^{(k)}, T_u^{(k)})$$

2. *The **graph embeddings** are uniformly upper Lipschitz w.r.t. $TMD^{(K)}$*

$$\|c^{\text{global}} - \hat{c}^{\text{global}}\|_p^p \leq C_2 \cdot TMD^{(K)}(G, \hat{G})^p$$

*Proof.* Since we are proving for uniformly Lipschitz, we will allow ourselves to omit the parameters for ease of notation, as the proof holds for any set of parameters.

**Proof of First Claim**   Let $u, v \in G$. The proof will be by induction on $k$. for $k = 0$, by definition

$$\|x_v^{(0)} - x_u^{(0)}\|_p^p = TD^p(T_v^{(0)}, T_u^{(0)})$$

We now assume correctness for $k - 1$ and prove for $k$.

1. First, using the properties of $\phi^{(k)}$ and the induction hypothesis we get:

$$\|\phi^{(k)}(\mathcal{N}_v^{(k-1)}) - \phi^{(k)}(\mathcal{N}_u^{(k-1)})\|_p^p \leq c_\phi W_1^p(x_{v,\bullet}^{(k-1)}, x_{u,\bullet}^{(k-1)})$$

$$\overset{C.2}{\leq} c_\phi \cdot c_1 W_p^p(x_{v,\bullet}^{(k-1)}, x_{u,\bullet}^{(k-1)}) = c_\phi \cdot c_1 (\min_{\tau \in S_n} \sum_{s=1}^n \|x_{v,s}^{(k-1)} - x_{u,\tau(s)}^{(k-1)}\|_p)^p$$

$$\overset{C.1}{\leq} c_\phi \cdot c_1 \cdot c_2 (\min_{\tau \in S_n} \sum_{s=1}^n \|x_{v,s}^{(k-1)} - x_{u,\tau(s)}^{(k-1)}\|_p^p)$$

$$\overset{\text{ind. hypothesis} + F.1}{\leq} c_\phi \cdot c_1 \cdot c_2 \cdot c_{(k-1)} \sum_{s=1}^n TD^p(T_{v,s}^{(k-1)}, T_{u,\tau^*(s)}^{(k-1)})$$

$$\overset{C.1}{\leq} c_\phi \cdot c_1 \cdot c_2 \cdot c_{(k-1)} \cdot c_3 (\sum_{s=1}^n TD(T_{v,s}^{(k-1)}, T_{u,\tau^*(s)}^{(k-1)}))^p$$

$$= c W_{TD,1}^p(T_{v,\bullet}^{(k-1)}, T_{u,\bullet}^{(k-1)})$$

Where $\tau^* := \text{argmin}_{\tau \in S_n} \sum_{s=1}^n TD(T_{v,s}^{(k-1)}, T_{u,\tau(s)}^{(k-1)})$, $c_1, c_2, c_3$ are the relevant constants from the lemmas we used and $c$ is the multiplication of all previous constants.

2. Next, we note that

$$TD(T_v^{(k-1)}, T_u^{(k-1)}) \leq TD(T_v^{(k)}, T_u^{(k)})$$

Since the depth $k - 1$ computation trees are subtrees of the depth $k$ computation trees.

3. Next, using the properties of $\psi^{(k)}$, the induction hypothesis and the above we get:

$$\|x_v^{(k)} - x_u^{(k)}\|_p^p = \|\psi^{(k)}(x_v^{(k-1)}, \phi^{(k)}(\mathcal{N}_v^{(k-1)})) - \psi^{(k)}(x_u^{(k-1)}, \phi^{(k)}(\mathcal{N}_u^{(k-1)}))\|_p^p$$

$$\leq c_\psi \cdot (\|x_v^{(k-1)} - x_u^{(k-1)}\|_p^p + \|\phi^{(k)}(\mathcal{N}_v^{(k-1)}) - \phi^{(k)}(\mathcal{N}_u^{(k-1)})\|_p^p)$$

$$\overset{\text{ind. hypothesis}}{\leq} c_\psi \cdot (c_{(k-1)}TD^p(T_v^{(k-1)}, T_u^{(k-1)}) + \|\phi^{(k)}(\mathcal{N}_v^{(k-1)}) - \phi^{(k)}(\mathcal{N}_u^{(k-1)})\|_p^p)$$

$$\overset{1}{\leq} c_\psi \cdot (c_{(k-1)}TD^p(T_v^{(k-1)}, T_u^{(k-1)}) + cW_{TD,1}^p(T_{v,\bullet}^{(k-1)}, T_{u,\bullet}^{(k-1)}))$$

$$\overset{2}{\leq} c_\psi \cdot (c_{(k-1)}TD^p(T_v^{(k)}, T_u^{(k)}) + cW_{TD,1}^p(T_{v,\bullet}^{(k-1)}, T_{u,\bullet}^{(k-1)}))$$

$$\overset{C.1}{\leq} c_\psi \cdot max(c_{(k-1)}, c) \cdot c_4 (TD(T_v^{(k)}, T_u^{(k)}) + W_{TD,1}(T_{v,\bullet}^{(k-1)}, T_{u,\bullet}^{(k-1)}))^p$$

$$\leq c_\psi \cdot max(c_{(k-1)}, c_1) \cdot c_4 \cdot 2^p \cdot TD^p(T_v^{(k)}, T_u^{(k)})$$

Where $c_4$ is the relevant constant from norm equivalence, concluding the proof of the first claim.

**Proof of Second Claim**  Let us denote the final message passing features corresponding to the nodes of G by $x_v^{(K)}$, and the depth-$K$ computation trees corresponding to each node of G by $T_v^{(K)}$. Similarly, we denote the features corresponding to the nodes of $\hat{G}$ by $\hat{x}_v^{(K)}$ and the trees by $\hat{T}_v^{(K)}$. Then, using the properties of $\eta$ and first part of the theorem we get

$$\|c^{\text{global}} - \hat{c}^{\text{global}}\|_p^p = \|\eta(x_\bullet^{(K)}) - \eta(\hat{x}_\bullet^{(K)})\|_p^p$$

$$\leq c_\eta W_1^p(x_\bullet^{(K)}, \hat{x}_\bullet^{(K)}) \overset{C.2}{\leq} c_\eta \cdot c_1 W_p^p(x_\bullet^{(K)}, \hat{x}_\bullet^{(K)}) = c_\eta \cdot c_1 (\min_{\tau \in S_n} \sum_{s=1}^n \|x_s^{(K)} - \hat{x}_{\tau(s)}^{(K)}\|_p^p)$$

$$\overset{(*)}{\leq} c_\eta \cdot c_1 \cdot C \cdot \sum_{s=1}^n TD^p(T_s^{(K)}, \hat{T}_{\tau^*(s)}^{(K)}) \overset{C.1}{\leq} c_\eta \cdot c_1 \cdot C \cdot c_2 \cdot (\sum_{s=1}^n TD(T_s^{(K)}, \hat{T}_{\tau^*(s)}^{(K)}))^p$$

$$= c_\eta \cdot c_1 \cdot C \cdot c_2 \cdot TMD^{(K)}(G, \hat{G})^p$$

Where (*) is the first claim and lemma F.1, $c_1, c_2$ are the relevant constants from norm equivalence and $\tau^* = \text{argmin}_{\tau \in S_n} \sum_{s=1}^n TD(T_s^{(K)}, \hat{T}_{\tau(s)}^{(K)})$, concluding the proof.

$\square$

### F.1 SORTMPNN

In the main text we stated the informal theorem

**Theorem 4.3.** *(informal) For any given* $W \geq 1, K \geq 0$*, SortMPNN with width* $W$ *and depth* $K$ *is lower Lipschitz in expectation with respect to* $TMD^{(K)}$.

We next state the theorem formally and present the proof.

Our results in this section hold for any number of MPNN iterations $K$, and any choice of 'widths' $W_1, \ldots, W_{K+1}$. For simplicity we prove this for the apriori hardest case, $W_1 = 1 = \ldots = W_{K+1}$. We will call this version of SortMPNN thin-SortMPNN.

Explicitly, thin-SortMPNN is defined by:

**SortMPNN: For k $= 1 \ldots, K$**

$$AGGREGATE: c_v^{(k)} = \text{sort}\left(a^{(k)} \cdot x_{v,\bullet}^{(k-1)}\right)$$

$$COMBINE: x_v^{(k)} = d^{(k)} \cdot concat(x_v^{(k-1)}, c_v^{(k)})$$

Note that we omit the second inner product in the definition of $S_z$ as it is superfluous: this inner-product would be subsumed by the following inner product with $d^{(k)}$ in the *COMBINE* function.

The READOUT function is given by

$$READOUT : c^{\text{global}} = b^{(K+1)} \cdot \text{sort}\left(a^{(K+1)} \cdot x_{\bullet}^{(K)}\right),$$

We will denote by $\theta^{(k)}$ the concatenation of all network parameters up to the creation of the node features $x_v^{(k)}$, where $\theta^{(0)}$ is just an 'empty vector'. We denote all network parameters, including those used by the readout function by $\theta^{(K+1)}$. The distribution on each of the parameter vectors $a^{(k)}, b^{(k)}, d^{(k)}$ is taken to be uniform on the unit sphere of the relevant dimension, as discussed in the main text.

We will now state our theorem on lower Lipschitzness of thin-SortMPNN.

**Theorem F.3.** *(lower Lipschitz SortMPNN, formal) Under the above assumptions F, and given $a^{(k)}, b^{(k)}$ are distributed uniformly on the appropriate unit sphere, then for thin-SortMPNN with depth $K$ the following holds:*

1. *The **node embeddings** after $k$ message passing layers are lower Lipschitz in expectation w.r.t. $TD$ on their depth $k$ computation trees*

$$\mathbb{E}_{\phi^{(k)}}[\|x_v^{(k)} - x_u^{(k)}\|_p^p] \geq c_1 \cdot TD^p(T_v^{(k)}, T_u^{(k)})$$

2. *The **graph embeddings** are lower Lipschitz in expectation w.r.t. $TMD^{(K)}$*

$$\mathbb{E}_{\phi^{(K+1)}}[\|c^{\text{global}} - \hat{c}^{\text{global}}\|_p^p] \geq c_2 \cdot TMD^{(K)}(G, \hat{G})^p$$

For the proof of the theorem we will need the following simple but useful lemma

**Lemma F.4.** *Let $D, N$ be natural numbers and $p > 1$. Then there exists a positive $\delta = \delta(D, N, p)$ such that, for all $N$ fixed vectors $x_1, \ldots, x_N$ in $\mathbb{R}^D$*

$$\mathbb{P}\left\{a \in S^{D-1}| \quad |a \cdot x_i| \geq \delta\|x_i\|_p, \forall i = 1, \ldots, N\right\} \geq 1/2$$

*Proof of Lemma F.4.* Due to equivalence of norms C.1, it is sufficient to prove the claim when $p = 2$.

For every $y \in S^{D-1}$ and $\delta > 0$, denote

$$B(y, \delta) = \{a \in S^{D-1}| \quad |a \cdot y| < \delta\}$$

We note that for any fixed positive $\delta$ and $y, y' \in S^{D-1}$, the probability of $B(y, \delta)$ and $B(y', \delta)$ will be the same, due to the rotation invariance of the uniform measure on $S^{D-1}$, and the fact that if $R$ is a rotation taking $y$ to $y'$, then

$$a \in B(y, \delta) \text{ iff } |a \cdot y| < \delta \text{ iff } |Ra \cdot Ry| < \delta \text{ iff } Ra \in B(y', \delta)$$

We can therefore denote the probability of $B(y, \delta)$ by $p_\delta$, and this definition does not depend on the choice of $y$. Next, note that $p_\delta$ is a non-negative seuence converging monotonely to $0$ as $\delta \to 0$. Accordingly,we can choose some $\delta_0$ such that $p_\delta < \frac{1}{2N}$.

Now, assume we are given $N$ fixed points in $\mathbb{R}^D$. Without loss of generality we can assume the first $M$ points are non-zero, and the last $N - M$ points are zero. We then have

$$\mathbb{P}\left\{a \in S^{D-1}| \quad |a \cdot x_i| \geq \delta\|x_i\|_2, \forall i = 1, \ldots, N\right\} = \mathbb{P}\left\{a \in S^{D-1}| \quad |a \cdot x_i| \geq \delta\|x_i\|_2, \forall i = 1, \ldots, M\right\}$$

$$= \mathbb{P}\left\{a \in S^{D-1}| \quad |a \cdot \frac{x_i}{\|x_i\|_2}| \geq \delta, \forall i = 1, \ldots, M\right\}$$

$$= 1 - \mathbb{P}\left(\cup_{i=1}^M B(\frac{x_i}{\|x_i\|_2}, \delta_0)\right)$$

$$\geq 1 - \sum_{i=1}^M \mathbb{P}\left(B(\frac{x_i}{\|x_i\|_2}, \delta_0)\right)$$

$$\geq 1 - \frac{M}{2N}$$

$$\geq 1 - \frac{N}{2N} = \frac{1}{2}.$$

$\square$

Using the lemma, we can now prove the theorem:

*Proof of Theorem F.3.* In general, to show that a function $f : X \times W \to Y$ is lower-Lipschitz in expectation, it suffices to show that there exist $\delta, \epsilon > 0$ such that for all $x_1, x_2 \in X$, the probability of the set $\{w \in W : \frac{|f(x_1,w)-f(x_2,w)|}{d_X(x_1,x_2)} > \epsilon\}$ is larger than $\delta$. We will use this alternative requirement in the proof of both parts of the theorem.

**Proof of First Claim** For ease of notation, we will use $W$ throughout this proof to denote $W_{TD,1}^{\bar{T}_z}$, and $W^p(\cdot, \cdot) = W_{TD,1}^{\bar{T}_z}(\cdot, \cdot)^p$.

We prove the claim be induction on $k$. For $k = 0$ we have equality

$$\|x_v^{(0)} - x_u^{(0)}\|^p = TD^p(T_v^0, T_u^0)$$

We now assume correctness for $k - 1$ and prove for $k$. Now note that

1. With probability of at least $p_{k-1}$ on $\theta^{(k-1)}$, we know that for all nodes $u, v$

$$\|x_v^{(k-1)} - x_u^{(k-1)}\|^p \geq c_k TD^p(T_v^{(k-1)}, T_u^{(k-1)})$$

2. Once $\theta^{(k-1)}$ is fixed, and all features $x_v^{(k-1)}$ are determined, we can use Lemma F.4 to show that with an appropriate $\delta > 0$ and probability of at least $1/2$ on $a^{(k)}$,

$$|a^{(k)} \cdot (x - y)| \geq \delta\|x - y\|$$

for all $x, y$ in the set $\{x_v^{(k)}\}_{v \in V} \cup \{z_k\}$. It follows that, with probability $1/2 p_{k-1}$ on $(a^{(k)}, \theta^{(k-1)})$

$$
\begin{aligned}
\|c_v^{(k)} - c_u^{(k)}\|_p^p &= \|\mathrm{sort}\left(a^{(k)} \cdot x_{v,\bullet}^{(k-1)}\right) - \mathrm{sort}\left(a^{(k)} \cdot x_{u,\bullet}^{(k-1)}\right)\|_p^p \\
&= \min_{\tau \in S_n} \sum_{s=1}^n |a^{(k)} \cdot (x_{v,s}^{(k-1)} - x_{u,\tau(s)}^{(k-1)})|^p \\
&\geq \min_{\tau \in S_n} \sum_{s=1}^n \delta^p \|x_{v,s}^{(k-1)} - x_{u,\tau(s)}^{(k-1)}\|_2^p \\
&\geq \delta^p C_1^p \min_{\tau \in S_n} \sum_{s=1}^n \|x_{v,s}^{(k-1)} - x_{u,\tau(s)}^{(k-1)}\|_p^p \\
&\overset{(*)}{\geq} \delta^p C_1^p \tilde{c}_k \min_{\tau \in S_n} TD^p(T_{v,s}^{(k-1)}, T_{u,\tau(s)}^{(k-1)}) \\
&\geq \delta^p C_1^p \tilde{c}_k C_2 \left[\min_{\tau \in S_n} TD(T_{v,s}^{(k-1)}, T_{u,\tau(s)}^{(k-1)})\right]^p \\
&= c_{k,1} W^p(\mathcal{T}_v^{(k-1)}, \mathcal{T}_u^{(k-1)})
\end{aligned}
$$

where in the last equation we used $c_{k,1}$ to denote the product of all constants appearing previously. (*) is the induction hypothesis and lemma F.1.

3. Once $(a^{(k)}, \theta^{(k-1)})$ are determined, we know that $c_v^{(k)}$ and $x_v^{(k-1)}$ are determined. Thus, for an appropriate positive $\delta'$, we know that with probability of at least $1/2$ on $d^{(k)}$, and thus

with probability of at least $p_k := p_{k-1}/4$ on $\theta^{(k)} = (d^{(k)}, a^{(k)}, \theta^{(k-1)})$, we have

$$
\begin{aligned}
|x_v^{(k)} - x_u^{(k)}|^p &= |d^{(k)} \cdot concat(x_v^{(k-1)} - x_u^{(k-1)}, c_v^{(k)} - c_u^{(k)})|^p \\
&\geq (\delta')^p \|concat(x_v^{(k-1)} - x_u^{(k-1)}, c_v^{(k)} - c_u^{(k)})\|_p^p \\
&\geq (\delta')^p \left( \|x_v^{(k-1)} - x_u^{(k-1)}\|_p^p + \|c_v^{(k)} - c_u^{(k)}\|_p^p \right) \\
&\geq (\delta')^p c_{k-1} TD^p(T_v^{(k-1)}, T_u^{(k-1)}) + (\delta')^p c_{k,1} W^p(\mathcal{T}_v^{(k-1)}, \mathcal{T}_u^{(k-1)}) \\
&\geq (\delta')^p \min\{c_{k-1}, c_{k,1}\} \|x_v^{(0)} - x_u^{(0)}\|_p^p + W^p(\mathcal{T}_v^{(k-1)}, \mathcal{T}_u^{(k-1)}) \\
&\overset{(*)}{\geq} C(\delta')^p \min\{c_{k-1}, c_{k,1}\} \left( \|x_v^{(0)} - x_u^{(0)}\|_p + W(\mathcal{T}_v^{(k-1)}, \mathcal{T}_u^{(k-1)}) \right)^p \\
&= c_k TD^p(T_v^{(k)}, T_u^{(k)})
\end{aligned}
$$

where in the last equation we used $c_k$ to denote all constants incurred up to this step. In the inequality (*) we used equivalence of norms in euclidean space, with an appropriate constant $C$.

This concludes the proof of the first part of the theorem.

**Proof of Second Claim** Let us denote the final message passing features corresponding to the nodes of $G$ by $x_v^{(K)}$, and the depth-K computation trees corresponding to each node of $G$ by $T_v^{(K)}$. Similarly, we denote the features corresponding to the nodes of $\hat{G}$ by $\hat{x}_v^{(K)}$ and the trees by $\hat{T}_v^{(K)}$. By applying the first part of the theorem (to the disjoint union of the graphs $G$ and $\hat{G}$), we know that with probablity of at least $p_K$ on the parameters $\theta_K$, we have that

$$
|x_v^{(K)} - \hat{x}_u^{(K)}|^p \geq c_k TD^p(T_v^{(K)}, \hat{T}_u^{(K)})
$$

Once $\theta_K$ is fixed, all node features $x_v^{(K)}$ and $\hat{x}_u^{(K)}$ are determined. By Lemma F.4, we have for an appropriate $\delta > 0$ that, with probablity of at least $1/2$ on $a^{(K+1)}$,

$$
|a^{(K+1)} \cdot (x_v^{(K)} - \hat{x}_u^{(K)})| \geq \delta \|x_v^{(K)} - \hat{x}_u^{(K)}\|_2, \quad \forall u, v \in [n].
$$

It follows that for an appropriate $\delta'$, with probability of at least $p_k/4$ on $\theta_{K+1} = (b^{(K+1)}, a^{(K+1)}, \theta_K)$

$$
\begin{aligned}
|c^{\text{global}} - \hat{c}^{\text{global}}|^p &= |b^{(K-1)} \cdot [\text{sort}\left(a^{(K+1)} \cdot x_\bullet^{(K)}\right) - \text{sort}\left(a^{(K+1)} \cdot \hat{x}_\bullet^{(K)}\right)]|^p \\
&\geq \delta'^p \|\text{sort}\left(a^{(K+1)} \cdot x_\bullet^{(K)}\right) - \text{sort}\left(a^{(K+1)} \cdot \hat{x}_\bullet^{(K)}\right)\|_p^p \\
&= \delta'^p \min_{\tau \in S_n} \sum_{v=1}^{n} |a^{(K+1)} \cdot x_v^{(K)} - a^{(K+1)} \cdot \hat{x}_{\tau(v)}^{(K)}|^p \\
&\geq (\delta \cdot \delta')^p \min_{\tau \in S_n} \sum_{v=1}^{n} \|x_v^{(K)} - x_{\tau(u)}^{(K)}\|_p^p \\
&\overset{(*)}{\geq} (C \cdot \delta \cdot \delta')^p \left( \min_{\tau \in S_n} \sum_{v=1}^{n} \|x_v^{(K)} - x_{\tau(u)}^{(K)}\|_p \right)^p \\
&\geq (C \cdot \delta \cdot \delta')^p TMD^{(K)}(G, \hat{G})
\end{aligned}
$$

where for $(*)$ we used equivalence of norms in Euclidean spaces, and the last inequality uses the first claim and lemma F.1.

$\square$

## F.2 MPNN WITH LOWER-HÖLDER COMPONENTS ISN'T NECESSARILY LOWER-HÖLDER

Consider the following setting: we consider graphs in $G_{\leq n}(\Omega)$ where $\Omega = [-2, 2] \subseteq \mathbb{R}$. Assume we have an MPNN $f$ with two layers, where the aggregation functions are the one dimensional

functions $q : (x_1, \ldots, x_n) \mapsto \sum_{i=1}^{n} \text{ReLU}(x_i - b)$ where $b \sim [-3, 3]$. We showed this type of multiset aggregation is Lower-Hölder in expectation C.4. We claim that with these one-dimensional aggregations and for any COMBINE function, there are graphs $G_1, G_2 \in G_{\leq n}(\Omega)$ which can be separated by two iterations of 1-WL, but cannot be separated by two of our MPNN iterations, for any choice of network parameters. An illustration of these graphs is given in Figure 5. The parameter $\epsilon$ can be taken to be any fixed number in $(0, 1/2)$. As can be seen, the depth-2 computation trees of the nodes at level 2 (children of the root, filled in red) differ between the graphs. Therefore, the 1-WL test will succeed in determining the graphs are non-isomorphic after two iterations.

We now explain why the aforementioned MPNN won't succeed in separating the graphs in two iterations. In our analysis, we use the names of the nodes defined in the figure: $a_1, \ldots, e_1$ for the nodes of $G_1$ and $a_2, \ldots, e_2$ for the nodes of $G_2$. We denote that feature vector at (e.g.,) $a_1$ after the $k$-th iteration by $a_1^{(k)}$.

The core reason for the failure of two iterations of the MPNN to separate $G_1, G_2$ is that, depending on the value of $b$, the first aggregation will not be able to simultaneously separate the multisets $\{\!\{-\epsilon, \epsilon\}\!\}$ and $\{\!\{0, 0\}\!\}$ and the multisets $\{\!\{1 - \epsilon, 1 + \epsilon\}\!\}$ and $\{\!\{1, 1\}\!\}$. In general, we can consider three options, depending on the value of $b$:

1. $b \in [-3, -\epsilon] \cup [\epsilon, 1 - \epsilon] \cup [1 + \epsilon, 3]$: In this case, the aggregation won't separate the multisets $\{\!\{-\epsilon, \epsilon\}\!\}$ and $\{\!\{0, 0\}\!\}$, and it also won't separate the multisets $\{\!\{1 - \epsilon, 1 + \epsilon\}\!\}$ and $\{\!\{1, 1\}\!\}$.

   In this case, after the first message passing layer, we will get equality between the corresponding features of nodes $a_1^{(1)} = a_2^{(1)} = c_1^{(1)} = c_2^{(1)}, b_1^{(1)} = b_2^{(1)} = d_1^{(1)} = d_2^{(1)}$. This means the multisets $\{\!\{a_1^{(1)}, b_1^{(1)}\}\!\}$ and $\{\!\{a_2^{(1)}, b_2^{(1)}\}\!\}$ will be equal, and so will the multisets $\{\!\{c_1^{(1)}, d_1^{(1)}\}\!\}$ and $\{\!\{c_2^{(1)}, d_2^{(1)}\}\!\}$. Therefore, the second message passing iteration will result in the equality between the corresponding features of nodes $e_1^{(2)} = e_2^{(2)}, f_1^{(2)} = f_2^{(2)}$, and the graphs won't be separated.

2. $b \in (-\epsilon, \epsilon)$: In this case the aggregation can separate the multisets $\{\!\{-\epsilon, \epsilon\}\!\}$ and $\{\!\{0, 0\}\!\}$, but not the multisets $\{\!\{1 - \epsilon, 1 + \epsilon\}\!\}$ and $\{\!\{1, 1\}\!\}$.

   In this case, after a single message passing layer, we will get equality between features of the nodes $a_1^{(1)} = a_2^{(1)}, c_1^{(1)} = c_2^{(1)}, d_1^{(1)} = b_2^{(1)} = b_1^{(1)} = d_2^{(1)}$.

   This means the multisets $\{\!\{a_1^{(1)}, b_1^{(1)}\}\!\}$ and $\{\!\{a_2^{(1)}, b_2^{(1)}\}\!\}$ will be equal, and so will the multisets $\{\!\{c_1^{(1)}, d_1^{(1)}\}\!\}$ and $\{\!\{c_2^{(1)}, d_2^{(1)}\}\!\}$. Therefore, the second message passing iteration will result in the equality between the corresponding features of nodes $e_1^{(2)} = f_2^{(2)}, f_1^{(2)} = e_2^{(2)}$, and the graphs won't be separated.

3. $b \in (1 - \epsilon, 1 + \epsilon)$: In this case the aggregation can separate the multisets $\{\!\{1 - \epsilon, 1 + \epsilon\}\!\}$ and $\{\!\{1, 1\}\!\}$, but not the multisets $\{\!\{-\epsilon, \epsilon\}\!\}$ and $\{\!\{0, 0\}\!\}$.

   This will lead to a similar result as the previous case, just with a slight change to the equivalence classes of the nodes.

Overall, we have shown that the proposed MPNN, which is composed of lower-Hölder in expectation components, fails to separate the above graphs for any choice of parameters. Therefore, it cannot be lower-Hölder in expectation w.r.t. $TMD^{(2)}$.

We note that the graphs can be separated by a similar ReLUMPNN which is wider, or has additional message passing layers. Our point here is just that there exists an MPNN with $K$ message passing layers, composed of lower-Hölder in expectation components, which isn't lower Hölder in expectation w.r.t. $TMD^K$.

### F.3 RELUMPNN AND SMOOTHMPNN

In the main text of the paper we stated the following informal theorem

**Theorem 4.2.** *Assume that ReluMPNN with depth $K$ is $\alpha$ lower-Hölder in expectation with respect to $TMD^{(K)}$, then $\alpha \geq 1 + \frac{K+1}{p}$. If SmoothMPNN with depth $K$ is $\alpha$ lower-Hölder in expectation then $\alpha \geq 2^{K+1}$.*

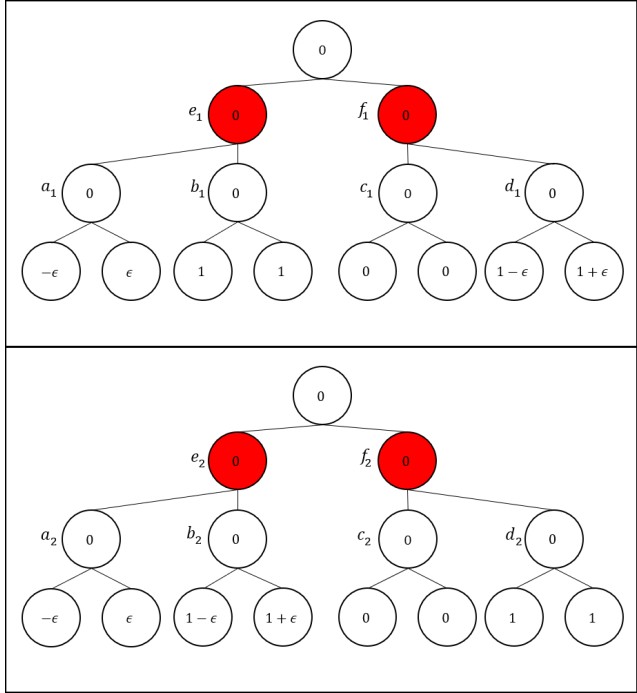

Figure 5: A pair of labeled graphs $G_1$ (top) and $G_2$ (bottom) used to prove that MPNNs with Hölder building blocks aren't necessarily Hölder

We note that the distribution on the parameters taken in this theorem is the same as described throughout the text. The aggregation used for both MPNNs at hand is of the form

$$\{\!\{x_1, \ldots, x_s\}\!\} \mapsto \sum_{i=1}^{s} \rho(a \cdot x_i - b)$$

where $a$ and $b$ are drawn uniformly from the unit sphere and some interval $[-B, B]$ respectively. Of course, in ReluMPNN $\rho = \mathrm{ReLU}$ while with smoothMPNN $\rho$ is a smooth function. The parameters of the linear COMBINE function are also selected uniformly in the unit sphere.

$\epsilon$-**Tree dataset**  The proof of the theorem is based on a family of adversarial example which we denote by $\epsilon$-trees. The recursive construction depends on two parameters: a real number $\epsilon$, and an integer $T = 0, 1, \ldots$. This will give us a pair of trees $(G, \hat{G})$ where $G = G(T, \epsilon), \hat{G} = \hat{G}(T, \epsilon)$.

The recursive construction of $\epsilon$ trees is depicted in Figure 6. In the first step $T = 0$, we define 'trees' $a_0, b_0, c_0, d_0$ which consist of a single node, with feature value of $\epsilon, -\epsilon, 2\epsilon$ and $-2\epsilon$, respectively. We connect $a_0, b_0$ via a new root with feature value 0 to create the tree $G_0 = G(T = 0, \epsilon)$, as depicted in the left hand side of Figure 6. We do the same thing with $c_0, d_0$ to obtain the tree $\hat{G}_0 = \hat{G}(T = 0, \epsilon)$.

We now proceed recursively, again as shown in Figure 6. Assuming that for a given $T$ we have defined the trees $a_T, b_T, c_T, d_T, G_T, \hat{G}_T$, we define the corresponding trees for $T + 1$ as shown in the figure:

For example, $a_{T+1}$ is constructed by joining together two copies of $a_T$ and two copies of $b_T$ at a common root with feature value 0 as shown in the figure on the right hand side. From the figure one can also infer the construction of $b_{T+1}, c_{T+1}, d_{T+1}$ from $a_T, b_T, c_T, c_T$. Finally, the tree $G_{T+1}$ is constructed by joining together $c_{T+1}, d_{T+1}$ while $\hat{G}_{T+1}$ is constructed by joining together $a_{T+1}, b_{T+1}$

*Proof of Theorem 4.2.* Denoting an MPNN of depth $T$ by $f_T(G, w)$, where $w$ denotes the network's parameters, our goal is to provide bounds of the form

$$\mathbb{E}_w \| f_T(G(T, \epsilon), w) - f_T(\hat{G}(T, \epsilon), w) \|^p \leq C\epsilon^{p\beta} \tag{7}$$

where $\beta$ depends on $T$ and the activation used, and on the other had show that $TMD^T(G(T, \epsilon), \hat{G}(T, \epsilon))$ scales linearly in $\epsilon$. This will prove that the MPNN cannot be $\alpha$-lower Hölder with any $\alpha < \beta$.

We can see that $TMD^T(G(T, \epsilon), \hat{G}(T, \epsilon))$ scales linearly in $\epsilon$ because the TMD is homogenous. Namely, given any $G = (V, E, X)$ and $\epsilon > 0$, let $\epsilon G = (V, E, \epsilon X)$. Then for any $T$ we have

$$TMD^T(\epsilon G, \epsilon \hat{G}) = \epsilon \cdot TMD^T(G, \hat{G}).$$

We now turn to estimate the exponent $\beta$ in (7), for smooth or ReLU activations, and different values of $T$. To avoid cluttered notation we will explain what occurs for low values $T = 0$ and $T = 1$ in detail, and outline the argument for larger $T$.

When $T = 0$, we apply an MPNN of depth $T = 0$ to the graphs $G_0, \hat{G}_0$, which means that we just apply a readout function to the initial features. This is exactly the $\pm\epsilon$ example discussed in the main text, and so (7) will hold with $\beta = (p + 1)/p$ with ReLU activations, and $\beta = 2$ with smooth activations.

We now consider the case $T = 1$. Here we apply we an MPNN of depth $T = 1$ to the graphs $G_1, \hat{G}_1$. Let us denote the nodes of $G_1$ by $V_1$, using the natural correspondence defined by the figure (middle) we can think of $V_1$ as the nodes of $\hat{G}_1$ as well. We denote the node feature values at node $v$ after a single MPNN iteration by $x_{v,w}$ and $\hat{x}_{v,w}$, where $w$ denotes the network parameters as before.

Note that for the root $r$ we have that $x_{r,w} = \hat{x}_{r,w}$ for any choice of parameters $w$.

At initialization, all leaves of the trees $G_0, G_1$ are assigned one of the values $\epsilon, -\epsilon, 2\epsilon, -2\epsilon$ (according to the labeling $a_0, b_0, c_0$ or $d_0$). If two leaves are assigned the same value (both are denoted by, say, $a_0$), then they will have the same node feature after a single message passing iteration, since they are only connected to their father, and all fathers have the same initial node feature.

We now consider the two remaining nodes. We denote the node connected to the root from the left by $v_1$ and the node connected to the root from the right by $v_2$.

Let us first focus on the smooth activation case. We note that the children of $v_1$ and $v_2$, in both graphs, have initial features of order $\epsilon$, and the all sum to the same value zero. By considering the Taylor approximation of the activation, as discussed in the proof of Theorem 3.3, we see that for an appropriate constant $C$ we have that for all small enough $\epsilon$,

$$|c_{v,w} - \hat{c}_{u,w}| \leq C\epsilon^2, \text{ for all parameters } w \tag{8}$$

where $v$ and $u$ could either be $v_1$ or $v_2$, and we recall that $c_{v,w}$ is the output of the aggregation function, applied to the initial features of the neighbors of $v$ in $G$ (and $\hat{c}_{u,w}$ is defined analogously for the neighbors of $u$ in $\hat{G}$).

Additionally, we note that for every parameter $w$ we have that

$$c_{v_1,w} + c_{v_2,w} = 2 \left[ \psi_w(\epsilon) + \psi_w(-\epsilon) + \psi_w(2\epsilon) + \psi_w(-2\epsilon) \right] = \hat{c}_{v_1,w} + \hat{c}_{v_2,w} \tag{9}$$

where $\psi_w$ denotes the neural network used for aggregation. The next step in the MPNN procedure is the COMBINE step, after which we will have a bound of the form (9), namely

$$|x_{v,w} - \hat{x}_{u,w}| \leq C\epsilon^2, \text{ for all parameters } w \tag{10}$$

since the COMBINE functions we use are uniformly Lipschitz, and we will also have or every parameter $w$ we have that

$$x_{v_1,w} + x_{v_2,w} = \hat{x}_{v_1,w} + \hat{x}_{v_2,w} \tag{11}$$

since our COMBINE function is linear.

Now, when applying the readout function, we have eleven node features in the multiset of nodes of $G_1$ and $\hat{G}_1$. For any parameter vector $w$, if we remove the nodes $v_1, v_2$, we get two identical multisets. Therefore, we only need to bound the difference

$$\| \left[ \eta_w(x_{v_1,w}) + \eta_w(x_{v_2,w}) \right] - \left[ \eta_w(\hat{x}_{v_1,w}) + \eta_w(\hat{x}_{v_2,w}) \right] \|$$

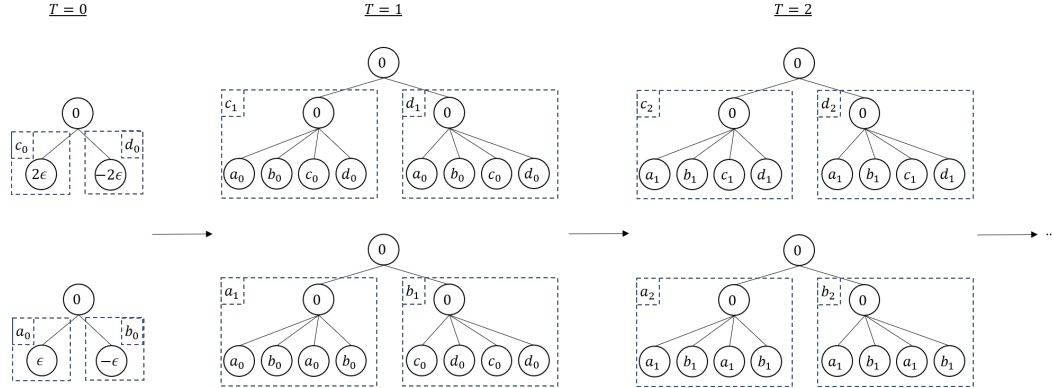

Figure 6: $\epsilon$-tree dataset construction: The trees are built using basic elements $a_t, b_t, c_t, d_t$. At step $T$, the basic elements are defined using subtrees from the previous step.

Since by (11) the features sum to the same value, and all are the same up to $\epsilon^2$, this difference goes like $\epsilon^4$. This is what we wanted. For general $T$, we will be able to apply this same argument recursively $T + 1$ times to get the lower bound of $2^{T+1}$ for the smooth exponent.

Now, let us consider what happens when we replace the smooth activation with ReLU activation. In This case too, the only features which really matter are those corresponding to $v_1, v_2$. Recalling our analysis of the $\pm\epsilon$ for ReLU activations, we note that if the bias of the network falls outside the interval $[-2\epsilon, 2\epsilon]$, then all node features $x_{v_1,w}, x_{v_2,w}, \hat{x}_{v_2,w}, \hat{x}_{v_2,w}$ will all be identical. If the bias does fall in that interval, then the difference between these node features can be bounded from above by $C\epsilon$ for an appropriate constant $C$. When applying readout, there is a probability of $\sim \epsilon^2$ that the final global features will be distinct (if the parameters of aggregation, and the parameters of readout, are both well behaved). The difference in this case can again be bounded by $C\epsilon$. It follows that

$$\mathbb{E}_w \| f_T(G(T, \epsilon), w) - f_T(\hat{G}(T, \epsilon), w) \|^p \le C\epsilon^2 \epsilon^p$$

which will lead to a lower bound on the Hölder constant $\alpha \ge (1 + \frac{2}{p})$.

For general $T$, we will get that there is a probability of $\sim \epsilon^{T+1}$ to obtain any separation, and that, if this separation occurs, the difference between features will be $\le C \cdot \epsilon$. As a result, the expectation

$$\mathbb{E}_w \| f_T(G(T, \epsilon), w) - f_T(\hat{G}(T, \epsilon), w) \|^p$$

will scale like $\epsilon^{T+1+p}$, which leads to our lower bound on the exponent for ReLU activations. $\qquad\square$

## G EXPERIMENTS

### G.1 HARDWARE

All experiments were executed on an a cluster of 8 nvidia A40 49GB GPUs.

### G.2 ARCHITECTURE VARIATIONS

In this subsection we will lay out the architectural variations that were considered for SortMPNN and AdaptMPNN, that were omitted from the main text for brevity. We include explanations of these variations with the connection between them and the theoretical analysis.

**SortMPNN** First off, we must choose how to model of the blank vector, since SortMPNN uses the blank vector explicitly in (2). We propose three options for the (*blank method*):

1. *Learnable*: For every multiset aggregation function there is a unique learnable blank vector that is used for augmentation. Since we assume the node features are from a compact subset

of $\mathbb{R}^d$, there exist valid augmentation vectors, and upon training we can expect the learnable vector to converge to a valid augmentation vector. Note that this method assumes training takes place.

2. *Iterative update*: In this method, the blank vector of each layer is the output of the previous MPNN layers on the blank tree, where the blank tree node feature is set to zero. This works under two assumptions: (1): The nodes in the dataset don't have the feature zero. (2): The intermediate layer outputs on computation trees from the data don't clash with the same layer output on the blank tree. This seems relatively reasonable since SortMPNN is bi-Lipschitz in expectation, which in expectation should lead to injectivity.

3. *Zero*: simply using the zero vector as the blank vector. This works under the assumption that the zero vector is neither a valid initial feature, nor a valid output of intermediate layers. This is the weakest of the three, but was experimented with nonetheless.

Given a multiset $X$, we experimented with two variations for the aggregation implementation. Both share step (1), but differ in step (2):

1. *projection*: $X \mapsto colsort(A\rho_{(z)}(X)) := Y \in \mathbb{R}^{m \times n}$, where $A \in \mathbb{R}^{m \times d}$.

2. *collapse*: In this step, we either perform (1) a *matrix* collapse: $rowsum(B \odot Y)$ where $\odot$ is the element-wise product and $B \in \mathbb{R}^{m \times n}$ or (2) a *vector* collapse: $Yb$, for $b \in \mathbb{R}^n$.

Note that when choosing to use the matrix collapse, under the assumption that the blank method provides a valid blank vector, this is equivalent to running $m$ separate copies of $S_z$ (2) with independent parameters (at least upon initialization), just like we proposed in the main text. When using the vector collapse, the aggregation takes the form of multiple instances of $S_z$ except that that the parameter $b$ used in the inner product is shared across instances. Note the Hölder expectation doesn't change in this case, but the variance is likely higher. We explored this option since it reduces the number of parameters needed for the aggregation, potentially easing the optimization process.

In addition, we experimented with adding bias to the projection, and to the vector collapse.

**AdaptMPNN**    The main detail we left out from the main text is the fact that the output of $m_{\text{ReLU}}^{\text{adapt}}$ is in $\mathbb{R}^4$, whereas we would want it to be in $\mathbb{R}$ in order to be able to stack $m$ instances and get an output in $\mathbb{R}^d$. In order to get the desired output dimension, we compose $m_{\text{ReLU}}^{\text{adapt}}$ with a projection, and the aggregation on a multiset $X \in \mathbb{R}^{d \times r}$ as follows:

1. *Stacked Adaptive ReLU*:

$$Y := \begin{bmatrix} m_{\text{ReLU}}^{\text{adapt}}(X; a_1, t_1) \\ \vdots \\ m_{\text{ReLU}}^{\text{adapt}}(X; a_m, t_m) \end{bmatrix} \in \mathbb{R}^{m \times 4}$$

2. *Project*: $rowsum(B \odot Y)$ where $\odot$ is the element-wise product and $B \in \mathbb{R}^{m \times 4}$ has rows that are drawn uniformly independently from $S^3$.

As we proved previously C.3, an inner product with a vector drawn uniformly from $S^{d-1}$ is bi-Hölder in expectation. From composition properties B.1, this means that the above stacking of multiple independent instances of $project \circ m_{\text{ReLU}}^{\text{adapt}}$ keeps the Hölder properties of $m_{\text{ReLU}}^{\text{adapt}}$.

In addition, we experimented with four optional changes to the aggregation:

1. Adding the sum $\sum a \cdot x_j$ a 5'th coordinate to the output of $m_{\text{ReLU}}^{\text{adapt}}$. The idea behind this choice is that the sum is an informative feature which is encountered when using the standard ReLU activations and bias smaller than the minimal feature in the multiset. For adaptive ReLU, this summation will only occur when $t = 0$.

2. Note that the adaptive relu uses the parameter $t \in [0, 1]$ to choose a bias within the minimal and maximal multiset values. When training, we experimented with either clamping $t$ to $[0, 1]$ or otherwise lifting this constraint.

3. We experimented with adding a bias term to the projection step, since this can potentially lead to stronger expressivity.

4. the initialization of $t$ was optionally chosen to be linspace between $[0, 1]$.

### G.3 ADVERSARIAL MULTISET DATASET EXPERIMENTS

Figure 1 and figure 2 both made use of the adversarial multiset pairs which are constructed as is described in detail in section C.2.

To produce figure 1, we ran $m_{relu}, m_{\sigma}$ on a single adversarial pair of multisets that have 8 scalar elements, with 7 equal moments between the multisets, with $\epsilon = 0.1$.

To produce figure 2, we used 200 adversarial pairs, each containing 16 scalar elements, where each pair shares the first 15 moments. The pairs differ in the value of $\epsilon$, where $\epsilon \in [0, 1]$

To further strengthen the evidence of the importance of expected Hölder stability, we performed two additional experiments, where the analyzed models were trained on adversarial datasets.

First, we trained the models on 4 different datasets created as described in C.2. The four datasets were constructed such that each consists of 100 pairs of graphs with $n - 1$ shared moments, where $n$ is one of 1,3,7,15 per dataset. The 100 pairs are constructed by creating the pair from C.2, and multiplying all the multiset elements by $\epsilon$, where $\epsilon \in np.linspace(0.01, 0.1, 100)$. In each pair, the multiset that started the iterative construction process with the values $\{\!\{-1, 1\}\!\}$ was assigned the label 0, and the other was assigned the label 1. Then, these 100 pairs were randomly split into train, validation and test (0.8/0.1/0.1). The per model test set accuracy results as a function of number of equal moments is presented in figure 7a. We see that the better the expected Hölder exponent (depicted for this type of dataset in figure 2), the better the post-training accuracy.

Since the first experiment makes use of a dataset that doesn't expose adaptive ReLU's worst-case behavior, we proceed to repeat the experiment, except that now make a slight change to the dataset multisets: for each multiset from the previous experiment, we add to the multisets two more elements with the values 2 and $-2$. This causes the adaptive capabilities of adaptive ReLU to become useless, since the largest and smallest values in the multiset are far larger than the values needed to separate the values that are proportional to $\epsilon$. The results are depicted in figure 7b.

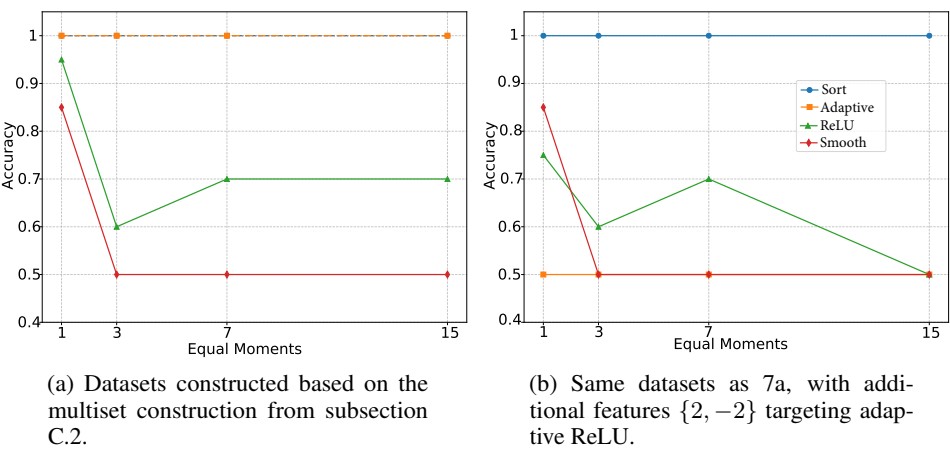

(a) Datasets constructed based on the multiset construction from subsection C.2.

(b) Same datasets as 7a, with additional features $\{2, -2\}$ targeting adaptive ReLU.

Figure 7: Test set accuracy for different configurations of datasets with equal moments.

### G.4 $\epsilon$-TREE DATASET

**Distortion experiment** The experiment depicted in 3 was run on a set of $1,000$ pairs of trees with $\epsilon$ going from 0.05 to 0.4. The first 500 trees are of height $h = 3$ and were input to MPNNs of depth $d = 1$, and the other 500 trees were of height $h = 4$ and were input to MPNNs of depth $d = 2$. The plots in figure 3 were separated by MPNN depth in order for the phenomenon to be clearly visible. We note that the feature of all non-leaf nodes which appear with 0 in figure 6, was set to

be 1. All the models in the experiment used linear combination as the combine function, had an embedding dimension of $45,000$, except SortMPNN that used an embedding dimension of $2,048$ due to resource constraints. The large width was used to show the behavior of the MPNNs as we approach the expected value over the parameters. In order to plot the curve in each subplot, the minimal ratio $r_{min} = l_2/TMD^\alpha$ was computed for each graph pair where $\alpha$ is the theoretical lower-Hölder exponent. Then, $r_{min} \cdot x^\alpha$ was plotted. Note that $r_{min}$ was only computed for half of the pairs with the largest TMD values in order to avoid the case where the embedding distance is zero leading to $r_{min} = 0$.

Throughout the experiment, double precision was used in order to capture the small differences as needed. The measured embedding distance was taken over the output of the readout function, without further processing.

**Separation experiment**   For the trained graph separation experiment, $100$ pairs of $\epsilon$ trees of height $h = 4$ with $\epsilon \in [0.1, 1]$ were used. Trees corresponding to the first row in figure 6 were given the label 0, and trees from the second row were given the label 1. The models were trained by inputting a single pair as a batch per training iteration. All models were composed of 2 message passing layers, followed by readout and a single linear layer to obtain a logit. All of $\{64, 128, 256, 512\}$ were tested for the embedding dimension with results consistent regardless of the embedding dimension. The models were trained for 10 epochs using the adamw optimizer and a learning rate of 0.01.

In an attempt to see the difference in quality between ReluMPNN and SmoothMPNN on this dataset, we reran the above, using the 'smallest' tree hight in our adversarial construction ($h = 3, d = 1$), and used a single message passing layer. However, even in this setting both ReluMPNN and SmoothMPNN completely failed to achieve separation.

### G.5   TUDATASET

Results on datasets that appear in table 3 for GIN, GCN, GraphSage are from (Xu et al., 2019), except for GIN on NCI109 which is from (Chuang & Jegelka, 2022). The reported results in 3 for the fully trained SortMPNN and AdaptMPNN were chosen out to be the best single result out of 30 random hyper-parameter choices from the following hyperparameters: batch size$\in \{32, 64\}$, depth$\in [2 - 5]$, embed dim$\in \{16, 32, 64\}$, combine$\in \{ConcatProject, LinearCombination, LTSum\}$, dropout$\in \{0, 0.1, 0.2\}$,output mlp depth $\in \{1, 2, 3\}$, weight decay $\in \{0, 0.01, 0.1\}$, lr$\in [0.0001, 0.01]$, optimizer $\in [adam, adamw]$, #epochs$= 500$. In addition, for SortMPNN the blank method was taken to be iterative update, and the collapse method was taken to be from {matrix, vector} without bias begin used at any stage. For AdaptMPNN, adding the sum of the multiset was a hyper-parameter, as was the choice if to clamp the parameter $t$ to be in $[0, 1]$. The choice of the 30 random configurations was chosen with wandb (Biewald, 2020) sweeps.

As stated in the main text, results are reported as in (Xu et al., 2019). Namely, we report the mean and standard deviation of the validation performance over stratified 10-folds, where the validation curves of all 10 folds are aggregated, and the best epoch is chosen.

### G.6   LRGB

In both experiments from table 4, the models adhere to the 500K parameter limit. The hyper-parameters were tuned for a single seed, and then the configuration with the best validation results was rerun for 4 different seeds. We report the mean and standard deviation on the test set in table 2. The code from the official LRGB github repo was used to run these experiments, with the only change being the addition of our architectures. The reported results for other models are taken from (Tönshoff et al., 2023).

Note that any theoretical result we presented that made use of a uniform distribution on the unit sphere holds also for any $t$-normalized unit sphere, i.e. $x\|\|x\| = t$, where $t \in R$, where only the multiplicative constants in the proofs will change. In this experiment, for SortMPNN we normalized the parameters sampled from the unit sphere as follows: (1) the parameter $a$ was sampled from a $\frac{1}{d}$-normalized sphere, (2) the parameter $b$ was sampled from a $\frac{1}{n}$-normalized sphere. This normalization was chosen through hyper-parameter tuning on the validation set.

**peptides-func** For SortMPNN, the following hyper-parameters were used to achieve the results shown in the table 4: blank method - learnable, collapse method - vector, bias used both in projection and collapse, combine - ConcatProject, embedding dimension - 165, message passing layers - 4, final mlp layers - 3, dropout - 0.05, weight decay - 0.01, positional encoding - RWSE (Dwivedi et al., 2022a), learning rate - 0.005, scheduler - cosine with warmup, optimizer - adamW. Number of parameters: 494865.

For AdaptMPNN - the following hyper-parameters were used: the bias parameter $t$ was clamped to stay within $[0, 1]$, the sum of a multiset was added as a 5'th coordinate, bias added in project, collapse method - LinearCombination, embedding dimension - 220, message passing layers - 6, final mlp layers - 3, dropout - 0.05,weight decay - 0.01, positional encoding - RWSE, learning rate - 0.001, scheduler - cosine with warmup, optimizer - adamW. Number of parameters: 480915.

**peptides-struct** For SortMPNN: blank method - learnable, collapse method - vector, bias not used both in projection or collapse, combine - LinearCombination, embedding dimension - 200, message passing layers - 6, final mlp layers - 3, dropout - 0, weight decay - 0.01, positional encoding - LapPE (Dwivedi & Bresson, 2021), learning rate - 0.005, scheduler - step, optimizer - adamW. Number of parameters: 493858.

For AdaptMPNN:the bias parameter $t$ was clamped to stay within $[0, 1]$, the sum of a multiset was added as a 5'th coordinate, bias added in project, collapse method - LinearCombination, embedding dimension - 220, message passing layers - 6, final mlp layers - 3, dropout - 0,weight decay - 0.1, positional encoding - LapPE, learning rate - 0.001, scheduler - step, optimizer - adamW. Number of parameters: 480973.

**small models** Additional experiments run on peptides struct were performed by running models with the exact same hyper-parameters aside from the width. Results are shown in figure 8. Exact width and number of parameters (including for the 500K budget) appear in table 6.

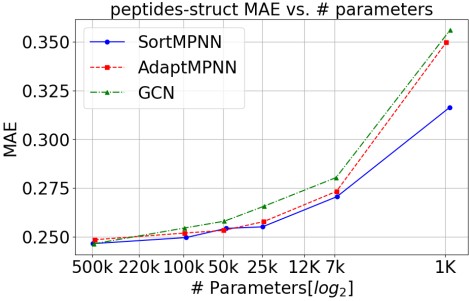

Figure 8: pep-struct, #params vs. MAE

Table 6: [width | #params] per model for each parameter budget

| Budget | SortMPNN | AdaptMPNN | GCN |
|---|---|---|---|
| 500K | [200 \| 493.8K] | [220 \| 478.8K] | [235 \| 488K] |
| 100K | [70 \| 95.9K] | [95 \| 98.6K] | [100 \| 98.4K] |
| 50K | [48 \| 47.3K] | [65 \| 49.3K] | [68 \| 48.8K] |
| 25K | [34 \| 24.9K] | [44 \| 24.3K] | [46 \| 24.2K] |
| 7K | [17 \| 6.7K] | [22 \| 6.7K] | [23 \| 6.8K] |
| 1K | [3 \| 922] | [5 \| 915] | [5 \| 982] |

## G.7 SUBGRAPH AGGREGATION NETWORKS

In this experiment, we made use of edge features. The incorporation of edge features was done via the concatenation of the edge and relevant node features, followed by a linear layer projecting the concatenated vector to the desired dimension. The projected vector was then passed through a ReLU activation.

Test set mean and standard deviation were computed over 10 random seeds. Hyper-parameters were chosen by randomly running 40 configurations with the DS edge deleted configuration, and choosing the best. the validation set was done using a single seed. The code provided with (Bevilacqua et al., 2022) was used to run these experiments, with the only change being the addition of our architectures.

**SortMPNN** Both for the DS and DSS experiments, the model consisted of 5 message passing layers, used sum Jumping Knowledge (JK) from (Xu et al., 2018), Combine - ConcatProject, matrix collapse, bias was added to projection, blank method - learnable. The model was trained for 400 epochs with lr=0.01 and batch size of 128.

**AdaptMPNN**    Both for the DS and DSS experiments, the model consisted of 5 message passing layers, sum JK for DS and last for DSS, combine - ConcatProject, sum wasn't added, $t$ was clapmed to $[0, 1]$, bias was added throughout aggregation steps. The model was trained for 400 epochs, with lr of 0.005 for DS and 0.01 for DSS.

## H    RUNTIME

| Model | GINE | GCN | SortMPNN | AdaptMPNN |
|---|---|---|---|---|
| Avg. time per epoch (s) | 12.16 | 14.25 | 16.08 | 26.77 |

Table 7: Average time per epoch of various models on the peptides-struct dataset. Average taken over 250 epochs.

