# OpenReview forum: "On the Hölder Stability of Multiset and Graph Neural Networks"
_ICLR.cc/2025/Conference — ICLR 2025 Oral_

### Official Review · Reviewer_TQxc · 2024-10-24

**Soundness:** 3
**Presentation:** 3
**Contribution:** 3
**Rating:** 8
**Confidence:** 3

**Summary:**

This paper proposed a pairwise separation quality analysis framework to study the Lipschitz and Hölder stability of functions on multisets and graphs. The authors proposed the notion of lower-Hölder in expectation (Definition 2.1), relaxing the uniform Hölder stability to the pairwise fashion. The authors showed that sum-based models on multisets and graphs are lower-Hölder in expectation with different exponents, depending on the activation choices (Table 1). Motivated from the theoretical analysis, the authors proposed AdaptMPNN and SortMPNN, which can separate the adversarial graph pairs and demonstrated improved empirical performance to existing MPNNs on standard benchmark datasets.

**Strengths:**

1. The study of separation power and stability of functions on multisets and graphs is of high interest to the community.

2. The proposed notion of lower-Hölder in expectation is interesting and can explain practical performance.

3. The proposed adversarial examples (i.e., $\pm \epsilon$-multisets, $\epsilon$-trees) are instructive and very useful to illustrative the theory.

**Weaknesses:**

1. The proposed SortMPNN deserves more remarks and discussion; see details in the Question section.

2. Empirical Comparison between AdaptMPNN, SortMPNN versus ReluMPNN and SmoothMPNN: While the authors compare these MPNN variants in Table 2 with the synthetic $\epsilon$-Tree dataset, the authors do not include the results of ReluMPNN and SmoothMPNN on the standard benchmark datasets (e.g. TUDataset, LRGB). Including such comparison could better bridge the theoretical insights in the paper to practice.

**Questions:**

1. The authors claimed that "SortMPNN is the first MPNN with both upper and lower Lipschitz guarantees" (Line 120-121). However, BLIS-Net proposed by Xu et al. [1] is shown to be Bi-Lipschitz with respect to a weighted inner product space. Can the authors compare and comment the differences?

2. Theorem 3.4 (SortMPNN is uniformly upper Lipschitz
and lower Lipschitz in expectation): This result seems straightforward given results in Balan et al. (2022). Can the authors discuss this more explicitly?

3. Connections between SortMPNNs and other related models: While sort-based aggregation function is rarely used in practice (and thus the novelty of the proposed SortMPNN), we can still see its "shades" in some related works. For example, in [2] the authors experimented with a sorting-based aggregation function; in [3] the authors proposed to learn a recursive aggregation function, which could in theory mimic a recursive sorting algorithm. Can the authors comment more of these related models?


References:

[1] Xu, Charles, et al. "BLIS-Net: Classifying and Analyzing Signals on Graphs." International Conference on Artificial Intelligence and Statistics. PMLR, 2024.


[2] Duvenaud, David K., et al. "Convolutional networks on graphs for learning molecular fingerprints." Advances in neural information processing systems 28 (2015).


[3] Ong, Euan, and Petar Veličković. "Learnable commutative monoids for graph neural networks." Learning on Graphs Conference. PMLR, 2022.

---

> ### Author Response · Authors · 2024-11-20
>
> Dear reviewer, thank you for the review! Below we address the questions you raised. We hope we were able to answer your concerns. Please let us know if you would like any further clarifications.
>
> Questions:
>
> >*The authors claimed that "SortMPNN is the first MPNN with both upper and lower Lipschitz guarantees" (Line 120-121). However, BLIS-Net proposed by Xu et al. [1] is shown to be Bi-Lipschitz with respect to a weighted inner product space. Can the authors compare and comment the differences?*
>
> This is a very interesting paper, thank you for pointing it out. We added the following to the related work, which we believe explains the differences:
> “In an additional recent work, \citet{xu2023blisnet} introduced a GNN designed to be bi-Lipschitz with respect to a weighted inner product space. Their approach, however, is limited to scenarios with a fixed graph topology, where only \textbf{scalar} node features vary. This fixed topology justifies the use of the weighted inner product space, which does not incorporate information about the graph structure.”
>
>
>
>
> >*Theorem 3.4 (SortMPNN is uniformly upper Lipschitz and lower Lipschitz in expectation): This result seems straightforward given results in Balan et al. (2022). Can the authors discuss this more explicitly?*
>
> Our bi-LIpschitz proof for *multisets*, namely Theorem 3.4, is indeed based on Balan’s work  as discussed directly after the statement of the theorem.
>
> Regarding SortMPNN which targets *graphs*: firstly, Balan did not discuss this scenario at all. Secondly, our proof does not employ ideas from Balan’s proof. In fact, the lower-Lipschitz in expectation bound we show holds even when the network has width 1, a fact which certainly does not hold w.r.t the standard notion of Lower-Lipschitz, since lower-lipschitz implies injectivity.
>
>
>
> >*Connections between SortMPNNs and other related models: While sort-based aggregation function is rarely used in practice (and thus the novelty of the proposed SortMPNN), we can still see its "shades" in some related works. For example, in [2] the authors experimented with a sorting-based aggregation function; in [3] the authors proposed to learn a recursive aggregation function, which could in theory mimic a recursive sorting algorithm. Can the authors comment more of these related models?*
>
> Thanks for suggesting these references. We already had a paragraph in the related work section describing other models for GNNs and multisets which used sorting. We added a reference to [2] in this section.
>
> *Weaknesses:*
> >*The proposed SortMPNN deserves more remarks and discussion; see details in the Question section.*
>
> We hope we were able to address your concerns. If there is something that is missing, please let us know.
>
>
> >*Empirical Comparison between AdaptMPNN, SortMPNN versus ReluMPNN and SmoothMPNN: While the authors compare these MPNN variants in Table 2 with the synthetic -Tree dataset, the authors do not include the results of ReluMPNN and SmoothMPNN on the standard benchmark datasets (e.g. TUDataset, LRGB). Including such comparison could better bridge the theoretical insights in the paper to practice.*
>
> The reason we didn’t include ReluMPNN and SmoothMPNN in the standard benchmarks is that we consider them as simplified models for  standard MPNN, which are based on similar building blocks. For example, the formulation of GIN is basically using such an aggregation. See corollary 6 and the paragraph after it in “How powerful are graph neural networks?".
> In our first experiment,  the $\epsilon$-tree experiment (Table 2), we see that indeed standard MPNN architectures behave similarly to ReluMPNN and SmoothMPNN in terms of Holder exponents. In the later experiments, our goal was to see whether the improved worst case stability of SortMPNN and AdaptMPNN, which we regard originally as our core models, translates into improved performance on real data.

---

> > ### Comment · Reviewer_TQxc · 2024-11-24
> >
> > Thank you for clarifying and addressing my comments in the updated manuscript. I also found the additional statement/proofs/experiments of the AdaptiveReLU model very interesting, which clearly strengthens the paper. I have raised my score and recommend acceptance of this work.

---

### Official Review · Reviewer_aRUR · 2024-10-31

**Soundness:** 3
**Presentation:** 3
**Contribution:** 3
**Rating:** 8
**Confidence:** 4

**Summary:**

The paper introduces a framework for quantifying the separation quality of multiset and graph neural networks (GNNs) via Hölder stability. This is motivated by recent observations that certain graph pairs, despite being theoretically distinguishable by Weisfeiler-Leman (WL)-expressive MPNNs (message passing GNNs), may have their embeddings "squashed" by GNNs, to the point that they are practically indistinguishable. The authors argue that strict lower Hölder continuity, which is only fulfilled for injective embeddings, is too restrictive as a concept and instead consider lower Hölder continuity in expectation over fixed parameter distributions. Starting with the multiset case, which appears as building blocks of MPNNs, they analyze the expected Hölder stability of popular architectures and introduce two new approaches, one based on bias scaling and the other on sorting, that improve the theoretical guarantees. They then employ these new multiset architectures in MPNNs and analyze their expected Hölder stabilities w.r.t. a WL distance, the tree mover's distance (TMD) (Chuang & Jegelka, 2022). Notably, the sort-based technique is shown to be lower Lipschitz in expectation w.r.t. the TMD. The new architectures AdaptMPNN and SortMPNN are empirically analyzed on toy and real-world datasets, demonstrating superior separation compared to popular MPNN baselines.

**Strengths:**

- The problem is well-motivated and I enjoyed reading the paper. To the best of my knowledge, the idea is novel, and as part of the ongoing effort to understand the geometry of MPNN embeddings, this work is a notable step.
- The theory is made quite accessible, and I particularly found the $\pm \epsilon$ example, which is used throughout the theory section, to be very instructive for a high-level understanding of the theorems.
- The presented empirical results are strong and insightful, showing that AdaptMPNN and SortMPNN can perfectly distinguish certain pairs of adversarial WL-distinguishable graphs, a task which standard MPNNs fail entirely.

**Weaknesses:**

In my opinion, the main weakness of the paper in its current state is that there is no theoretical analysis of the (expected) lower-Hölder properties of AdaptMPNN. While AdaptMPNN is one of the main methodological contributions of the work, only the lower-Hölder stability of the two "baselines" ReluMPNN and SmoothMPNN as well as SortMPNN are analyzed in section 4. Ideally, an outline of a worst case analysis as mentioned at the end of section 4, or at least some justification if or why there is something inherently harder about the analysis of AdaptMPNN compared to ReluMPNN might be an insightful addition to the manuscript.

**Questions:**

- Regarding the weaknesses: Are there any major difficulties in a Hölder stability analysis of AdaptMPNN compared to ReluMPNN? If so, a short paragraph explaining this might be interesting.
- I am a bit puzzled about the results for the adaptive ReLU architectures: Both in Figure 2 and Figure 3, the corresponding architectures seem to be almost perfectly lower Lipschitz on the data, behaving similarly to the sorted architectures. If I am not mistaken, however, this is not guaranteed by the theory that is derived. I am assuming that this is not a consequence of the bound from Theorem 3.2 not being tight, since it is stated in lines 308-309 that the adaptive ReLU architecture for multisets is *not* lower Lipschitz in expectation, as can be shown by some adversarial graphs. However, I couldn't find any further justification for this statement in the manuscript, as Theorem 3.2 only shows that $(p+1)/p$-lower Hölder continuity in expectation (which evaluates to $\alpha=3/2$ for the $\ell^2$ case), while Theorem 3.1 only refers to standard ReLU summation. Could you maybe point me to a proof/justification or add one, and generally comment a bit more on the performance of the adaptive ReLU methods in Figure 2 and 3?
- As also mentioned in the paper (lines 344-346), there are multiple WL distances that have been suggested so far. As expected lower-Hölderness also crucially depends on the chosen distance functions, is there any particular reason why you chose the TMD?
- An interesting question would be to analyze more quantitatively how the expected Hölder stability evolves over training, as the theory is only laid out for the initial parameters (and depends crucially on the initialization distributions). While the results of Table 2 suggest that the stability is preserved during training, I would find more quantitative empirical results concerning the evolvement of the Hölder stability with the NN weights during training very interesting to see. However, as this is very open-ended and potentially costly, I would like to emphasize that this is merely as a suggestion.

---

> ### Author Response · Authors · 2024-11-20
>
> Dear reviewer, thank you for the review! Below we address the questions you raised. We hope we were able to answer your concerns. Please let us know if you would like any further clarifications.
>
> Questions:
>
> >*I am a bit puzzled about the results for the adaptive ReLU architectures: Both in Figure 2 and Figure 3, the corresponding architectures seem to be almost perfectly lower Lipschitz on the data, behaving similarly to the sorted architectures. If I am not mistaken, however, this is not guaranteed by the theory that is derived. I am assuming that this is not a consequence of the bound from Theorem 3.2 not being tight, since it is stated in lines 308-309 that the adaptive ReLU architecture for multisets is not lower Lipschitz in expectation, as can be shown by some adversarial graphs. However, I couldn't find any further justification for this statement in the manuscript, as Theorem 3.2 only shows that -lower Hölder continuity in expectation (which evaluates to  for the  case), while Theorem 3.1 only refers to standard ReLU summation. Could you maybe point me to a proof/justification or add one, and generally comment a bit more on the performance of the adaptive ReLU methods in Figure 2 and 3?*
>
>
> We agree with these remarks, thanks. To address this we made the following improvements in the revised version of the submission:
>
> 1. We added to Theorem 3.2 the statement that the exponent  of (p+1)/p is tight, and updated the proof accordingly.
>
> 2. In Remark C.6 we explained intuitively why adaptive ReLU performs in a bi-Lipschitz like fashion in the adversarial examples used for Figures 2 and Figure 3. We also explained how to modify these examples so that the Holder behavior of adaptive ReLU is apparent.
>
> 3. In Figure 7 which we now added to the paper, we show how adaptive ReLU, which performs well on our vanilla adversarial example Figure 7(a), fails once they are modified as we suggest in Figure 7(b).
>
> In general, the idea is as follows: ReLU-summation can separate multisets {$-\epsilon,\epsilon$} and {$-2\epsilon,2\epsilon$} only when the bias is in the range $(-2\epsilon,2\epsilon\) $. The probability of the bias falling in this range is small, and this causes the non-lower-Lipschitzness of ReLU-Summation.
>
> The idea behind Adapt-MPNN is to choose the bias to lie between maximal and minimal features. In the example discussed above, this will automatically lead to the bias lying in the correct regime where it can separate the two multisets. Thus, on this example, Adapt-MPNN will behave like a lower-Lipschitz function, in expectation. However, if we add the same large positive and negative feature to both multisets, to obtain {$-1,1,-\epsilon,\epsilon$} and {$-1,1,-2\epsilon,2\epsilon$}, then the bias will be selected in the regime $(-1,1)$, and so the probability of the bias lying in the correct regime will be very low as in the standard ReLU case, and we will get a behavior consistent with the worst case (p+1)/p exponent.
>
> *Regarding the weaknesses:*
> >*Are there any major difficulties in a Hölder stability analysis of AdaptMPNN compared to ReluMPNN? If so, a short paragraph explaining this might be interesting.*
>
> We believe the same ideas used to prove Theorem 4.2 for ReluMPNN will apply to AdaptMPNN, but this would require modifying the epsilon tree example by adding leave to the trees with fixed features valued 1 and -1, similar to what is discussed above for the multisets.
>
> >*As also mentioned in the paper (lines 344-346), there are multiple WL distances that have been suggested so far. As expected lower-Hölderness also crucially depends on the chosen distance functions, is there any particular reason why you chose the TMD?*
>
> We believe there are several points presented in the original TMD paper that indicate its relevance for MPNN analysis: (1) Table 1 in their paper shows that a simple SVM based on TMD is competitive on some graph benchmarks. (2) Figure 4 in their paper shows the nice correlation between TMD and MPNN embeddings (in contrast to another WL-metric they compare to).
>
> In any case, we conjecture that many WL-metrics are equivalent (in a “bi-Lipschitz” sense). In research we are currently working on we have encouraging preliminary results regarding this conjecture.
>
>
>
> >*An interesting question would be to analyze more quantitatively how the expected Hölder stability evolves over training… However, as this is very open-ended and potentially costly, I would like to emphasize that this is merely as a suggestion.*
>
> This is definitely an interesting experiment. However, due to time and resource limitations we don’t believe we will be able to add this within the discussion period. We hope to address this in future work.

---

> > ### Comment · Reviewer_aRUR · 2024-11-23
> > **Response to Rebuttal**
> >
> > I thank the authors for their thoughtful rebuttal and for answering my questions. I particularly want to acknowledge that my concerns about AdaptMPNN were thoroughly addressed by the authors. As such, I am happy to raise my score. I encourage the authors to further work on the minor presentation issues in the work, as suggested by reviewer oS9G.

---

### Official Review · Reviewer_oS9G · 2024-10-31

**Soundness:** 4
**Presentation:** 2
**Contribution:** 3
**Rating:** 8
**Confidence:** 3

**Summary:**

This work analyzes the abillity for multiset and graph neural networks to separate inputs. Due to the non-injectivity of ReLU networks, the authors introduce the property of a network being Holder in expectation over random weights (Holderness is just an extension of Lipschitz continuity with an exponential factor $\alpha$). The authors then derive the lower-Holder exponents for a variety of models, and provide preliminary empirical evidence that lower Holderness is related to network performance on some synthetic tasks.

**Strengths:**

Overall, this is a heavily theoretical work that I am not well-equipped to confidently review. I have given it my best effort, although I will likely defer to other reviewers' judgements.

1. I really appreciate the $\epsilon$-tree experiment. I originally had my doubts that an architecture being $\alpha$ lower-Holder in expectation over *random weight parameters* is that meaningful for the practical ability of networks to distinguish inputs. Random-weight networks hardly reflect how trained networks ultimately behave. This experiment provides examples of networks which are and aren't $\alpha$ lower-Holder in expectation and shows that this property is critical in a synthetic task.
2. The theory is extensive but generally well-presented and the headline results are clear (although I did not check the proofs in detail). Proof sketches are included where appropriate.
3. The considered problem is interesting, and I appreciate how the authors generalized their framework to both multiset analysis framework to graph neural networks.
4. It's a surprising result to me that relu sum networks are $\alpha=3/2$ lower-Holder in expectation. I would have originally guessed that they would be just $\alpha=1$ lower-Holder (Lipschitz) in expectation.

**Weaknesses:**

1. Some of the experimental benefits are pretty marginal (especially in table 4, where the gains are smaller than the exerimental variance).
2. While the $\epsilon$-Tree experiment provides some evidence for the importance of $\alpha$ lower-Holderness in expectation over random weights, it just includes a few anecdotal examples of architectures which do and don't work. A really convincing experiment would consist of a scalar-parameterized collection of function classes (which are in turn parameterized). Say this scalar smoothly varied the resulting $\alpha$; I'd then hope to see that as $\alpha$ increased, performance increases.
3. The paper seems rushed, and has many typos. Consider lines 145-147 for example (there are many more such mistakes). The authors should thoroughly proofread their work.
4. Related, the paper presentation is somewhat haphazard, with many text-wrapped figures and tables; these really should be combined appropriately and top-aligned to the page with no text wrapping.

**Questions:**

Comments and questions

1. Definition 2.1 depends on choices of $\alpha$ and $p \geq 1$. Shouldn't we be talking about whether functions are $\alpha,p$ lower-Holder in expectation instead of just $\alpha$ lower-Holder in expectation?
2. The remarks on line 162-165 don't make sense to me. Consider: "Firstly, we note that if the set of $w$ for which $f (x; w)$ is $\alpha$ lower-Hölder has positive probability, then $f(x; w)$ will be $\alpha$ lower-Hölder in expectation." I don't see why this is true. If we take $p=1$ and move $c^p$ in Definition 2.1 to the denominator of the RHS, then the $w$ for which $f(x; w)$ is $\alpha$ lower-Holder produce a value inside the expectation which is $\geq 1$. Then all we can say is that the RHS expectation is atleast the probability mass of such $w$'s that satisfy the original lower-Holder condition -- this bound cannot even be greater than $1$, which is what the LHS would become after dividing by $c^p$.
3. Is the implication of the experiments in Figure 3 that the presented bounds are tight?
4. Does the adaptive ReLU activation provide any experimental benefits?
5. Am I correct in concluding from Theorem 3.3 that any smooth activation (e.g. sigmoid) cannot be $\alpha$ lower-Holder in expectation (otherwise $\alpha$ would have to be infinity)? This seems counterintuitive; consider an activation like $a^{-1} \log(1 + e^{ax})$. This function is smooth and thus cannot be $\alpha$ lower-Holder in expectation, but in the limit $a \to \infty$ approaches the ReLU function which is $\alpha=3/2$ lower-Holder in expectation.

Minor remarks:
* Reading just theorem 3.1, it is unclear how $a$ and $b$ are related to the rest of the thoerem. I suggest writing $m_{ReLU}(\ \cdot\ ; a,b)$ in the theorem statement.
* Figures should be exported as pdfs or pgfs in order to avoid text blurring.
* There is a line of work on the abillity of GNNs to discriminate signals from a spectral theory perspective which might be worth discussing:

[a] Gama, Fernando, Joan Bruna, and Alejandro Ribeiro. "Stability properties of graph neural networks." IEEE Transactions on Signal Processing 68 (2020): 5680-5695.

[b] Pfrommer, Samuel, Alejandro Ribeiro, and Fernando Gama. "Discriminability of single-layer graph neural networks." ICASSP 2021-2021 IEEE International Conference on Acoustics, Speech and Signal Processing (ICASSP). IEEE, 2021.

---

> ### Author Response · Authors · 2024-11-20
>
> Dear reviewer, thank you for the review! Below we address the questions you raised. We hope we were able to answer your concerns. Please let us know if you would like any further clarifications.
>
> Questions:
>
>
> >*Definition 2.1 depends on choices of $\alpha$ and p>=1. Shouldn't we be talking about whether functions are $\alpha,p$ lower-Holder in expectation instead of just lower-Holder in expectation?*
>
>
>
> You are correct. Indeed for ReLU-summation the lower Holder exponent depends on p. Nonetheless we choose to drop p from the notation to avoid cumbersome notation.
>
>
>
>
> >*regarding the comment starting with “the remarks on line 162-165…”*
>
> What we mean is that it will be lower-Holder in expectation with the same exponent, but not necessarily with the same constant. Specifically, what you are saying is correct, the constant can be “corrected” by multiplying with the probability mass of said w’s (to the power of p). If you feel this is confusing we would happily add a clarification.
>
>
>
> >*Is the implication of the experiments in Figure 3 that the presented bounds are tight?*
>
> As we write in 428-434, figure 3 mainly illustrates that the expected lower Holder exponents of ReluMPNN and SmoothMPNN behave at least as bad as our analysis dictates: the exponent grows at least as the lower bound provided in table 1 (this lower bound is illustrated by the blue solid line). Note that Figure 3 is evaluated on the adversarial examples used to prove these lower bounds. It is certainly possible that there are more extreme adversarial examples, which we are not yet aware of, that will lead to even worse behavior.
>
> >*Does the adaptive ReLU activation provide any experimental benefits?*
>
>
> The main benefits of adaptive ReLU are theoretical. However, it provides good results on the synthetic $\epsilon$-trees (table 2), and competitive results on the datasets from TUDatasets (table 3) and peptides-func (table 4).
>
> >*Am I correct in concluding from Theorem 3.3 that any smooth activation (e.g. sigmoid) cannot be  lower-Holder in expectation (otherwise  would have to be infinity)? This seems counterintuitive; consider an activation like…  This function is smooth and thus cannot be  lower-Holder in expectation, but in the limit approaches the ReLU function which is  lower-Holder in expectation.*
>
> Our analysis is conducted on spaces of multisets with at most n elements. In this setting, Theorem 3.3 doesn’t imply that a smooth activation will result in a non lower-Holder in expectation parametric function, but only that if the function is in fact $\alpha$ lower-Holder in expectation, then $\alpha \ge n$ where $n$ is the maximal size of a multiset. However, your argument seems correct if one were to consider multisets with unbounded cardinality.
>
>  Regarding the provided example, it indeed can be used to show that the lower-Holder in expectation property doesn’t behave “continuously” with respect to limits in function spaces. This is true also for the standard lower-Holder property.
>
> To see this, we can look at the function $f_{\epsilon}(x)=x^2-\epsilon\cdot x$ on $[0,1]$ with $\epsilon\in [0,1]$. When $\epsilon\rightarrow 0$ the function $f_{\epsilon}$ converges to the function $ x^2$, which is lower-Holder with an exponent of $2$:
> $|x^2-y^2|=|x-y||x+y|\geq |x-y|^2$. However, $f_\epsilon$ is not injective on this domain, as it assigns $x=0$ and $x=\epsilon$ the same value. Therefore, it is not $\alpha$ lower Holder for any $\alpha$.
>
>
>
> *regarding “Minor Remarks”*
>
> Thanks! We changed the theorem statement as you suggested, and added the works you mentioned to the related work section.

---

> ### Author Response · Authors · 2024-11-20
> **Part 2 of rebuttal**
>
> Weaknesses:
>
> >*While the epsilon Tree experiment provides some evidence for the importance of $\alpha$ lower-Holderness in expectation over random weights, it just includes a few anecdotal examples of architectures which do and don't work. A really convincing experiment would consist of a scalar-parameterized collection of function classes (which are in turn parameterized). Say this scalar smoothly varied the resulting $\alpha$ ; I'd then hope to see that as $\alpha$  increased, performance increases.*
>
> Thanks for this suggestion. We took two steps to address this concern:
>
> 1. In an attempt to see the difference in quality between ReluMPNN and SmoothMPNN on the epsilon-tree
> dataset, we reran the above, using the 'smallest' tree height in our adversarial construction ($h=3,d=1$ as opposed to $h=4,d=2$ shown in the main text), and used a single message passing layer. However, even in this setting both ReluMPNN and SmoothMPNN completely failed to achieve separation. We added a paragraph discussing this in appendix G.4. This is consistent with our theory, since even when $d=1$ ReLU MPNN has a lower Holder exponent of at least 2, and SmoothMPNN has a much higher exponent. In contrast, SortMPNN which is bi-Lipschitz (exponent of 1), achieves perfect accuracy on these tasks.
>
> 2. We also added a new experiment: we took pairs of multisets with n elements and n-1 identical moments, and used them to construct a binary classification task. We then examined the success of the various multiset architectures as a function of n. In this scenario, the sort based function is bi-Lipschtiz, the ReLU-summation and adaptive-ReLU have an exponent of 3/2, and the smooth-summation multiset functions have a high exponent of at least n. The results we obtain are again consistent with our theory, where sortMPNN achieves perfect performance,  ReLU-summation and adaptive-ReLU outperforming smoothMPNN, and the performance of smoothMPNN deteriorating as n increases. This experiment is visualized in Figure 7 and described in Appendix G.3.
>
>
> *Typos...*
>
> Sorry about this. We fixed lines 145-147 and made an additional effort to fix typos throughout the text.
>
>
>
> *Regarding text-wrapped figures and tables*
>
> This was done due to the constraint on the number of pages. To solve this issue,  we moved  a figure into the appendix (in the revised version this is figure 8), which allowed us to top-align tables 3 and 4. We hope this makes the presentation more friendly.

---

> > ### Comment · Reviewer_oS9G · 2024-11-21
> >
> > Thank you for your thorough response and revisions. After reading through other reviewer comments and discussions, I think this work would be a good fit for ICLR and am raising my score. A few suggestions for the potential camera ready version:
> >
> > * Right now, I don't see mention of the additional experiments in G.3 in the main body of the paper. A one-sentence summary of the important takeaway could be good to include.
> >
> > * I appreciate that the authors justified Tables 3 and 4 -- I think some more combining is possible and would benefit the presentation. For example, Figure 1 & Table 1 could be combined into one top-aligned float, Figure 2 could be four horizontal subplots and then be top aligned, Tables 2 & 5 could be combined, and Figure 3 could be squashed vertically and stretched horizontally. Just a suggestion; I know this figure finagling is work, but it provides much additional polish that reflects the overall effort clearly put into the work.
> >
> > * The fonts in the figures are sans-serif -- I'd suggest using a serif font to match the body and exporting to pdf.
> >
> > * Capitalize e in "embedding distance" in Figure 3. Also write out Tree Mover's Distance on the x axis.

---

> > > ### Author Response · Authors · 2024-11-23
> > >
> > > Thank you :)
> > >
> > > Good point about G.3, we'll add this.
> > >
> > > We will indeed work on further polishing the presentation, we appreciate the suggestions!

---

### Official Review · Reviewer_gx6Y · 2024-11-02

**Soundness:** 3
**Presentation:** 3
**Contribution:** 3
**Rating:** 8
**Confidence:** 3

**Summary:**

The authors study stability and separability properties of multi-set and graph neural networks. They do this by establishing Holder lower bounds on the networks, which give some guarantees on separability. They argue that uniform Holder lower bounds are too restrictive, and instead propose a notion of lower Holder in expectation, which is averaged over a probability distribution on the parameters of the network. The main results in the paper are bounds on the lower Holder exponent for different types of graph and multiset neural networks. They find that common architectures have poor exponents and separability properties and propose two new message passing neural networks with better separation properties. They provide a number of numerical experiments with their newly proposed architectures, and compare against existing methods.

Some typos I noticed: "decays rapidly with the network’s depth .", "paces. , let W be some set of parameters", "function. we say that", "multisets (Zaheer et al., 2017) ,as well as MPNNs"

**Strengths:**

Deep graph neural networks is an important area of research, and there are an enormous number of different architectures that have been proposed. This paper provides some valuable insights into which networks give better separation properties. The lower Holder estimates in expectation are novel contributions to the literature, as far as I am aware. The new MPNN architectures with better separation properties are a nice contribution.

**Weaknesses:**

The lower bounds may be pessimistic, relying on pathological adversarial examples. It is not clear how practical the advances are, in that the worst case graphs may not be encountered in practice.

**Questions:**

Holder continuity is usually used for exponents in the range [0,1]. Functions that are Holder continuous with exponents larger than 1 are trivial constant functions. It is not clear to me why the main results with \alpha > 1 are interesting. Is it because it is only the lower bound and not the upper?

The choice of the probability measure under which Holder in expectation is taken seems to be essential. However, I don't see this appearing in the discussion or even in the proof sketches of most of the results. This was confusing to me. Even in Euclidean space, it seems it would be challenging to choose the right measure. If the data satisfied the manifold-hypothesis, for example, then a function that well-separates the data could be constant almost everywhere with respect to Lebesgue measure and thus have no lower bound with Lebesgue measure, but would have some appropriate lower bound with a measure tuned to the data.

In the definition of the Wasserstein metric on line 197 there is a p-norm on the right hand side, but it is W^1. Is this correct?

The authors give a lot of pathological adversarial examples of graphs that are difficult to separate. How practical are these examples?

---

> ### Author Response · Authors · 2024-11-20
>
> Dear reviewer, thank you for the review! Below we address the questions you raised. We hope we were able to answer your concerns. Please let us know if you would like any further clarifications.
>
> Questions:
>
> >*Holder continuity is usually used for exponents in the range [0,1]. Functions that are Holder continuous with exponents larger than 1 are trivial constant functions. It is not clear to me why the main results with \alpha > 1 are interesting. Is it because it is only the lower bound and not the upper?*
>
> Indeed, you are correct. The term Holder continuity usually refers to the upper bound, in which case an exponent greater than 1 indicates a constant function. However, our main results regard the lower bound, in which case the interesting exponents are $[1,\infty)$. A simple way to see this is by noting that for every function $f$ that is $\alpha$ *upper*-Holder, the inverse function $f^{-1}$ is $1/\alpha$ *lower*-Holder: let $s=f(x), t=f(y)$, then $|f^{-1}(s)-f^{-1}(t)|=|x-y|\ge |f(x)-f(y)|^{1/\alpha}=|s-t|^{1/\alpha}$.
>
>
>
> >*The choice of the probability measure under which Holder in expectation is taken seems to be essential. However, I don't see this appearing in the discussion or even in the proof sketches of most of the results. This was confusing to me. Even in Euclidean space, it seems it would be challenging to choose the right measure. If the data satisfied the manifold-hypothesis, for example, then a function that well-separates the data could be constant almost everywhere with respect to Lebesgue measure and thus have no lower bound with Lebesgue measure, but would have some appropriate lower bound with a measure tuned to the data.*
>
>
> we hope we understand the question correctly, but could it be you misunderstood the expectation to be taken over the data? When defining the notion of Holder in expectation in 2.1, we state that the expectation is taken over some distribution over the *parameters*, whereas the analysis should hold for *any pair* of fixed data points. This means the analysis we provide is a worst-case analysis with respect to the data, as is further explained in lines 167-172. We specify the parameter distribution for each function in the respective theorem. For example, in theorem 3.1 we state that $a$ is distributed uniformly on the unit sphere, and $b$ is distributed uniformly on the interval $[-B,B]$ (also recall our notation for uniformly distributed defined in Section 1.0.1)
>
> >*In the definition of the Wasserstein metric on line 197 there is a p-norm on the right hand side, but it is W^1. Is this correct?*
>
>
> Thanks for catching this mistake. The p-norm was changed to the 1-norm in the revised version of the submission.
>
> >*The authors give a lot of pathological adversarial examples of graphs that are difficult to separate. How practical are these examples?*
>
>
> This is a good question. Given we are interested in constructing an architecture that is well-behaved for any dataset, for which we don't know the data distribution ahead of time, we resorted to the worst-case analysis. Accordingly, we expect architectures with better expected Holder exponents to be more “robust”.
> Moreover, our experiments imply that good expected exponents often translate to good performance on real datasets.
>
> General:
> Thank you for catching the typos, we fixed these.

---

> > ### Comment · Reviewer_gx6Y · 2024-11-27
> >
> > Thanks for the additional clarifying details, and the revisions to the paper. I have revised my score higher.

---

### Author Response · Authors · 2024-11-20
**General comments to all reviewers**

Dear reviewers, we thank you all for dedicating time to provide us with high quality reviews. Aside from addressing your concerns personally, please note that any changes we made due to the review (aside from fixing typos) appear in the revised pdf in blue text to make finding the changes easier.

The main changes we made were:
We added to the statement and proof of  Theorem 3.2 the fact that the (p+1)/p lower exponent of adaptive ReLU is tight.
 We discuss in Remark C.6 why adaptive ReLU displays bi-Lipschitz-like behavior in many of our experiments.
We added Figure 7, which (a) shows more evidence that high exponents lead to inferior learning, on adversarial tasks and (b) shows an adversarial example where adaptive-ReLU fails.

---

### Meta-Review · Area_Chair_atRQ · 2024-12-10

**Metareview:**

This paper studies the separation power and stability of functions on graphs and multisets. In addition to the theoretical stability bounds, which hold in expectation, the paper provides explicit adversarial examples that illustrate the boundary cases of the theoretical results. It also empirically shows the connection between the Holder stability bounds and the model performance. The paper finds that common architectures have poor Holder exponents and separability properties and proposes two new message-passing neural networks with better separation properties All reviewers stated that this is a good paper.

**Additional Comments On Reviewer Discussion:**

The authors answered the concerns raised by the reviewers. All reviewers raised their score after the discussion period.

---

### Decision · Program_Chairs · 2025-01-22

Accept (Oral)